# Modulation of flight and feeding behaviours requires presynaptic IP$_3$Rs in dopaminergic neurons

**Anamika Sharma, Gaiti Hasan\***

National Centre for Biological Sciences, TIFR, Bangalore, India

**Abstract** Innate behaviours, although robust and hard wired, rely on modulation of neuronal circuits, for eliciting an appropriate response according to internal states and external cues. *Drosophila* flight is one such innate behaviour that is modulated by intracellular calcium release through inositol 1,4,5-trisphosphate receptors (IP$_3$Rs). Cellular mechanism(s) by which IP$_3$Rs modulate neuronal function for specific behaviours remain speculative, in vertebrates and invertebrates. To address this, we generated an inducible dominant negative form of the IP$_3$R (IP$_3$R$^{DN}$). Flies with neuronal expression of IP$_3$R$^{DN}$ exhibit flight deficits. Expression of IP$_3$R$^{DN}$ helped identify key flight-modulating dopaminergic neurons with axonal projections in the mushroom body. Flies with attenuated IP$_3$Rs in these presynaptic dopaminergic neurons exhibit shortened flight bouts and a disinterest in seeking food, accompanied by reduced excitability and dopamine release upon cholinergic stimulation. Our findings suggest that the same neural circuit modulates the drive for food search and for undertaking longer flight bouts.

**\*For correspondence:**
gaiti@ncbs.res.in

**Competing interests:** The authors declare that no competing interests exist.

## Introduction

The Inositol-1, 4, 5-trisphosphate receptor (IP$_3$R) is an Endoplasmic Reticulum (ER) resident ligand-gated calcium (Ca$^{2+}$) channel found in metazoans. In neurons, the IP$_3$R is activated through multiple classes of signaling molecules that include neuromodulators such as neuropeptides, neurotransmitters, and neurohormones. Studies from ex vivo vertebrate neurons have identified a role for the IP$_3$R in multiple cellular processes including regulation of neurite growth (*Takei et al., 1998*; *Xiang et al., 2002*), synaptic plasticity (*Fujii et al., 2000*; *Nishiyama et al., 2000*) and more recently pre-synaptic neurotransmitter release (*Gomez et al., 2020*). However, the relevance of IP$_3$-mediated signaling mechanisms to cellular processes and subsequent behavioural and neurophysiological outputs need better understanding. In non-excitable cells Ca$^{2+}$ release through the IP$_3$R regulates a range of cellular events including growth, secretion (*Inaba et al., 2014*), gene expression (*Ouyang et al., 2014*), and mitochondrial function (*Cárdenas et al., 2010*; *Bartok et al., 2019*). In excitable cells such as neurons, identifying cellular functions of the IP$_3$R is more complex because in addition to IP$_3$-mediated Ca$^{2+}$ release, there exist several plasma-membrane localised ion channels that bring in extracellular Ca$^{2+}$ in response to neurotransmitters and changes in membrane excitability.

Three IP$_3$R isotypes, IP$_3$R1, 2 and 3 are encoded by mammalian genomes (*Furuichi, 1994*; *Taylor et al., 1999*). Of these, IP$_3$R1 is the most prevalent isoform in neurons and is relevant in the context of several neurodegenerative disorders (*Terry et al., 2018*). Human mutations in IP$_3$R1 cause Spinocerebellar ataxia 15 (SCA15), SCA29 and Gillespie syndrome (*Hasan and Sharma, 2020*). Disease causing IP$_3$R1 mutations span different domains but several are clustered in the amino terminal IP$_3$ binding region, from where they impact ER-Ca$^{2+}$ release when tested in mammalian cell lines (*Ando et al., 2018*). A common feature of these neurological disorders is loss of motor-coordination or ataxia. The ataxic symptoms arise primarily from malfunction and/or degeneration of cerebellar Purkinje neurons where the IP$_3$R1 is expressed abundantly in the soma, dendrites, and axons.

Dendritic expression of the IP$_3$R in Purkinje neurons determines Long Term Depression (LTD), a form of post-synaptic plasticity (*Miyata et al., 2000*). Interestingly, a recent study in *Drosophila* also identified the IP$_3$R as an essential component of post-synaptic plasticity, required for decoding the temporal order of a sensory cue and a reward stimulus, in neurons of a higher brain centre, the Mushroom Body (*Handler et al., 2019*). Somatic expression of the IP$_3$R very likely contributes to maintenance of cellular Ca$^{2+}$ homeostasis (*Berridge, 2016*). A pre-synaptic role for the IP$_3$R has been demonstrated in the *Drosophila* neuromuscular junction (*Shakiryanova et al., 2011*), motor neurons (*Klose et al., 2010*) and most recently in axonal projections of Purkinje neurons (*Gomez et al., 2020*). Physiological and behavioural significance of pre-synaptic IP$_3$/Ca$^{2+}$ signals in either case, however, remain speculative.

To understand how IP$_3$R alters neuronal function and related neurophysiology and behaviour, we have in the past studied several mutants for the single IP$_3$R gene (*itpr*) in *Drosophila melanogaster* (*Joshi et al., 2004*). The *Drosophila* IP$_3$R shares 60% sequence identity, similar domain organisation, biophysical and functional properties with mammalian IP$_3$Rs (*Chakraborty and Hasan, 2012*; *Srikanth and Hasan, 2004*). Several hypomorphic and heteroallelic combinations of *Drosophila* IP$_3$R mutants are viable and exhibit flight deficits ranging from mild to strong, the focus of which lies in aminergic neurons (*Banerjee et al., 2004*). However, such mutants do not easily allow cell specific attenuation of IP$_3$R function. In this context, a dominant-negative mutant form of mammalian IP$_3$R1, generated recently, abrogated IP$_3$R function in mammalian cell lines (*Alzayady et al., 2016*). The mammalian dominant negative IP$_3$R1 gene was based on the finding that all monomers in the IP$_3$R tetramer need to bind IP$_3$ for channel opening (*Alzayady et al., 2016*). Failure of a single IP$_3$R monomer to bind IP$_3$ renders the resultant IP$_3$R tetramer non-functional. *Drosophila* IP$_3$Rs are also tetrameric suggesting that a similar strategy for generating a dominant-negative IP$_3$R could be used for cell-specific studies of neuronal function and behaviour. Consequently, we generated and characterised a *Drosophila* IP$_3$R dominant negative (IP$_3$R$^{DN}$) transgene. Cell-specific expression of *Drosophila* IP$_3$R$^{DN}$ allowed the identification of two pairs of flight modulating dopaminergic neurons. Neuromodulatory signals received by these neurons stimulate IP$_3$/Ca$^{2+}$ signals to regulate critical aspects of pre-synaptic cellular physiology with significant impact on flight and feeding behaviour.

## Results

### Ex vivo characterisation of a dominant negative IP$_3$R

To understand how Ca$^{2+}$ release through IP$_3$R affects cellular properties of neurons, we designed a mutant *itpr* cDNA to function as a dominant negative upon overexpression in wild-type *Drosophila* neurons. The dominant negative construct (*Itpr$^{DN}$*) was designed based on previous studies in mammalian IP$_3$Rs (*Figure 1A*; *Alzayady et al., 2016*). Three conserved basic residues in the ligand binding domain (R272, K531, and Q533) of the *Drosophila* IP$_3$R cDNA were mutated to Glutamine (Q) and the resultant mutant cDNA was used to generate GAL4/UAS inducible transgenic strains (*UASItpr$^{DN}$*) as described in Materials and methods. Expression from the *Itpr$^{DN}$* construct was validated by western blots of adult fly head lysates. IP$_3$R levels were significantly enhanced in fly heads with *Itpr$^{DN}$* as compared to genetic controls and were equivalent to overexpression of a wild-type IP$_3$R transgene (*Itpr$^+$*; *Figure 1B*, *Figure 1—source data 1*). IP$_3$-mediated calcium release from the IP$_3$R in the presence of *Itpr$^{DN}$*, was tested on previously characterised glutamatergic interneurons in the larval ventral ganglion known to respond to the muscarinic acetylcholine receptor (mAChR) ligand Carbachol. These neurons can be marked with a GAL4 strain (*vGlut$^{VGN6341}$*; *Jayakumar et al., 2016*; *Jayakumar et al., 2018*). Changes in cytosolic calcium were measured by visualising Ca$^{2+}$ dependent fluorescence changes of a genetically encoded Ca$^{2+}$ sensor GCaMP6m (*Chen et al., 2013*) in ex vivo preparations. *vGlut$^{VGN6341}$* marked neurons expressing IP$_3$R$^{DN}$ exhibit reduced as well as delayed Ca$^{2+}$ responses to Carbachol stimulation as compared to controls (*Figure 1C*). Ca$^{2+}$-released from the IP$_3$R is also taken up by mitochondria (*Bartok et al., 2019*). Hence we measured mitochondrial Ca$^{2+}$ uptake post Carbachol stimulation using mitoGCaMP, a mitochondrial targeted fluorescence sensor for Ca$^{2+}$ (*Lutas et al., 2012*). Similar to cytosolic GCaMP, attenuated Ca$^{2+}$ responses, that took longer to reach peak values, were observed upon carbachol stimulation in presence of IP$_3$R$^{DN}$ (*Figure 1D*). We attribute the residual Ca$^{2+}$ release observed in the cytosol and mitochondria to the persistence of some IP$_3$R tetramers with all four wild-type subunits, encoded by the

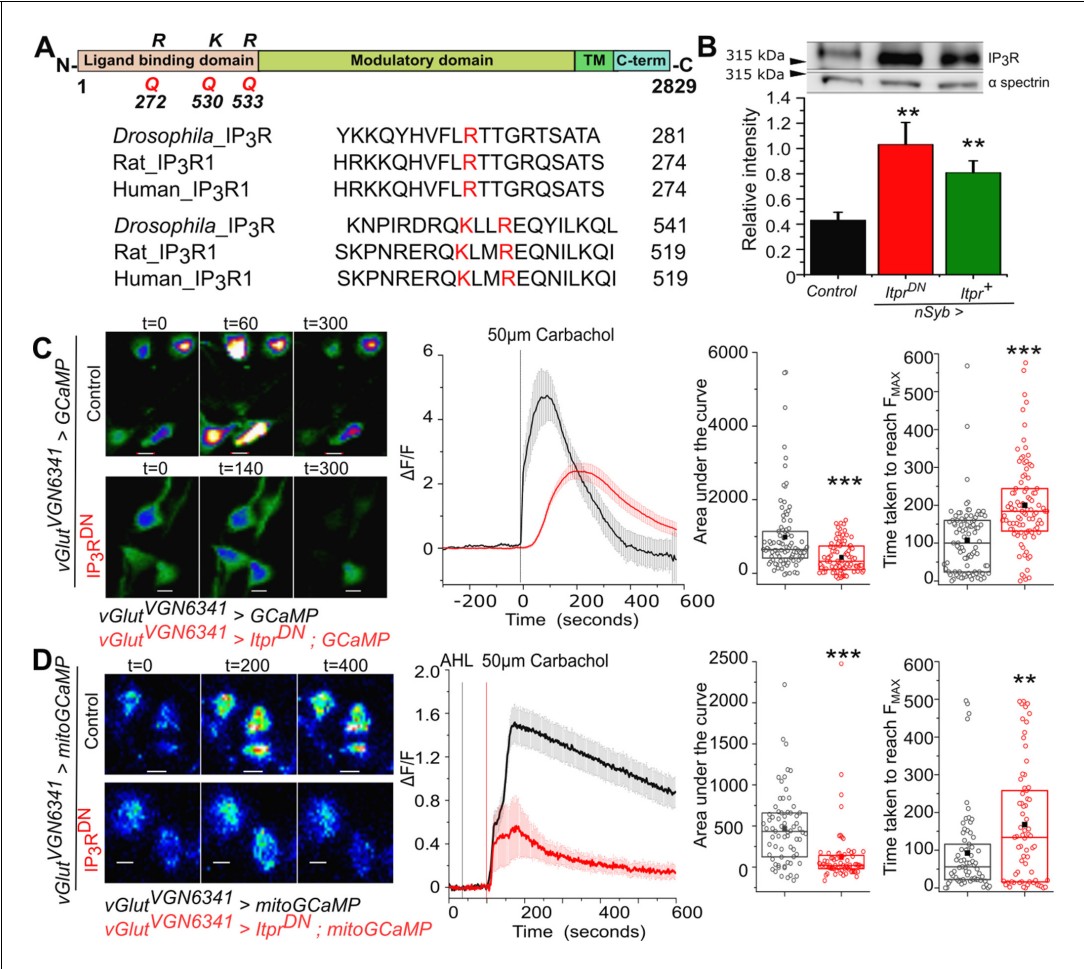

**Figure 1.** Generation and characterisation of a dominant negative IP₃R (IP₃R$^{DN}$). (**A**) Domain organisation of the *Drosophila* IP₃R. The Arginine (R) and Lysine (K) residues mutated to Glutamine (Q) to generate a dominant-negative IP₃R (IP₃R$^{DN}$) are shown (top). Alignment of the *Drosophila* IP₃R with Rat IP₃R1 and Human IP₃R1 in the region of the mutated amino acids. All three residues (red) are conserved (below). (**B**) Significantly higher immuno-reactivity against the IP₃R is observed upon pan-neuronal (*nsybGAL4*) expression of IP₃R$^{DN}$ (*UASitpr$^{DN}$*) and IP₃R (*UASitpr$^{+}$*) in adult heads. Statistical comparison was made with respect to *Canton S* as control, n=5, **p< 0.05, t-test. (**C**) Representative images show changes in cytosolic Ca$^{2+}$ in larval neurons of the indicated genotypes as judged by GCaMP6m fluorescence at the indicated time intervals after stimulation with Carbachol (Scale bars indicate 5 μm). Warmer colors denote increase in [Ca$^{2+}$]$_{cyt}$. Mean traces of normalized GCaMP6m fluorescence in response to Carbachol (4s/frame) in *vGlut$^{VGN6341}$GAL4* marked neurons, with (red) and without (black) IP₃R$^{DN}$ (ΔF/F ± SEM) (middle panel). Area under the curve (right) was quantified from 0 - 420s from the traces in the middle. Time taken to reach peak fluorescence (F$_{MAX}$) for individual cells (far right). *vGlut$^{VGN6341}$GAL4 > UAS GCaMP* N = 6 brains, 92 cells; *vGlut$^{VGN6341}$GAL4 > UAS Itpr$^{DN}$; UAS GCaMP* N = 6 brains, 95 cells. ***p< 0.005 (Two tailed Mann-Whitney U test). (**D**) Representative images show changes in mitochondrial Ca$^{2+}$ in larval neurons of the indicated genotypes as judged by mitoGCaMP6m fluorescence at the indicated time intervals after stimulation with Carbachol (Scale bars indicate 5 μm). Warmer colors denote increase in [Ca$^{2+}$]$_{cyt}$. Mean traces of normalized mitoGCaMP6m fluorescence in response to Carbachol (2s/frame) in *vGlut$^{VGN6341}$GAL4* marked neurons, with (red) and without (black) IP₃R$^{DN}$ (ΔF/F ± SEM) (middle panel). Area under the curve (right) was quantified from 100 to 400s from the traces in the middle. Time taken to reach peak fluorescence (F$_{MAX}$) for individual cells (far right). *vGlut$^{VGN6341}$GAL4 > UAS mitoGCaMP* N = 5 brains, 69 cells; *vGlut$^{VGN6341}$GAL4 > UAS Itpr$^{DN}$; UAS mitoGCaMP* N = 5 brains, 68 cells. ***p< 0.005, **p< 0.05 (Two tailed Mann-Whitney U test).

The online version of this article includes the following source data for figure 1:

**Source data 1.** Western Blots for IP₃R upon pan-neuronal expression of IP₃R$^{DN}$ and IP₃R WT in adult heads.

native *itpr* gene. In summary, both cytosolic and mitochondrial Ca$^{2+}$ measurements confirmed the efficacy of the *Drosophila* IP₃R$^{DN}$ for attenuating IP₃-mediated Ca$^{2+}$ release from ER stores upon GPCR stimulation in *Drosophila* neurons.

# The IP$_3$R is required in a subset of central dopaminergic neurons for maintenance of *Drosophila* flight

Next we tested the functional efficacy of *Itpr$^{DN}$* for attenuating neuronal function in *Drosophila*. From previous reports, we know that *itpr* mutants are flightless and their flight deficit can be rescued partially by overexpression of a wild-type cDNA construct (*Itpr$^+$*; *Venkatesh et al., 2001*) in mono-aminergic neurons (*Banerjee et al., 2004*). Subsequent studies identified mild flight deficits upon RNAi mediated knock down of the IP$_3$R in dopaminergic neurons (DANs) (*Pathak et al., 2015*) when tethered flight was tested for 30 s. Knock-down of the IP$_3$R in serotonergic neurons, that form another major subset of the tested monoaminergic neurons, did not give a flight deficit (*Sadaf et al., 2012*). Here we tested if pan-neuronal expression of the IP$_3$R$^{DN}$ affects longer flight bout durations in a modified tethered flight assay lasting for 15 min (see Materials and methods and *Manjila and Hasan, 2018*). Flies with pan-neuronal *nsybGAL4*-driven expression of *Itpr$^{DN}$* exhibit significantly reduced flight bouts (281.4 ± 38.9 s) as compared to the appropriate genetic controls *Itpr$^{DN}$/+* (773.3 ± 30.4 s) and *nsyb/+* (670.6 ± 41.3 s) (*Figure 2—figure supplement 1A*, *Figure 2— source data 4*). Stronger deficits in flight bout durations (185.2 ± 33.7 s) were obtained by expression of *Itpr$^{DN}$* in Tyrosine Hydroxylase expressing cells, that include a majority of dopaminergic neurons (*THGAL4*; *Friggi-Grelin et al., 2003*), as well as in a dopaminergic neuron subset (260 ± 30.5 s) marked by *THD'GAL4* (*Liu et al., 2012*; *Figure 2A and B*, *Figure 2—source data 1*). As additional controls the same GAL4 drivers were tested with overexpression of *Itpr$^+$*. Interestingly, flight deficits were also observed upon overexpression of *Itpr$^+$* across all neurons (*nsyb >Itpr$^+$*; 410 ± 31.6 s; *Figure 2—figure supplement 1A*), all TH-expressing cells (*TH >Itpr$^+$*; 482.2 ± 41.4 s) and a dopaminergic neuron subset, (*THD'>Itpr$^+$*; 433.5 ± 35.6 s; *Figure 2A*), although these were milder than with expression of *Itpr$^{DN}$*.

To confirm that shorter flight bouts in flies expressing *Itpr$^{DN}$* are a consequence of reduced IP$_3$R function in dopaminergic neurons and not an overexpression artefact, we expressed a previously validated *Itpr* RNAi (*Agrawal et al., 2010*) and recorded flight durations in an identical flight assay. Significantly reduced flight bouts, comparable to the flight deficits obtained upon expression of *Itpr$^{DN}$*, were observed by knockdown of the IP$_3$R with *nsybGAL4*, *THGAL4*, and *THD'GAL4* (*Figure 2A*, *Figure 2—figure supplement 1A*, *Figure 2—video 1*). These data confirm that IP$_3$R function is necessary in the *THD'* marked subset of dopaminergic neurons for maintenance of flight bouts. Further, it suggests that either decrease (*Itpr$^{DN}$* and *Itpr RNAi*) or increase (*Itpr$^+$*) of IP$_3$-mediated Ca$^{2+}$ release in *THD'* neurons affects flight bout durations.

As an independent test of IP$_3$R requirement, the wild-type IP$_3$R was overexpressed in dopaminergic neurons of an adult viable IP$_3$R mutant, *itpr$^{ka1091/ug3}$* or *itpr$^{ku}$* (*Joshi et al., 2004*). Overexpression of the IP$_3$R in monoaminergic neurons of *itpr* mutants can rescue free flight measured for short durations of 5–10 secs (*Banerjee et al., 2004*). Short (30 s, *Figure 2—figure supplement 1B*, *Figure 2— source data 5*) and long (900 s; *Figure 2C*, *Figure 2—source data 2*) flight bouts were measured in a heteroallelic viable *itpr* mutant combination *itpr$^{ku}$*, with *Itpr$^+$* overexpression in all neurons or in dopaminergic neurons and dopaminergic neuronal subsets. Pan-neuronal overexpression of the IP$_3$R (*nsyb >Itpr$^+$*) rescued short flight bouts partially in 10 out of 30 flies tested (*Figure 2—figure supplement 1B*). Interestingly, complete rescue of short flight was observed in flies rescued by IP$_3$R overexpression in dopaminergic neurons and their TH-D subsets but not the TH-C' subset (*Figure 2— figure supplement 1B*). Better rescue by IP$_3$R overexpression in dopaminergic neurons suggests weak expression of *nSybGAL4* in dopaminergic neurons, although this idea needs further verification. When tested for longer flight bouts, a partial rescue from dopaminergic neurons and TH-D subsets was observed (304.2 ± 25.1 s; *TH >Itpr$^+$*) as compared to control flies (747.8 ± 19.4 s; *Figure 2C*), suggesting that the IP$_3$R regulates flight bout durations from both dopaminergic and certain non-dopaminergic neurons (*Agrawal et al., 2010*). The *TH-D* and *TH-C GAL4*s express in anatomically distinct central dopaminergic neurons of which the *THD1GAL4* and *THD'GAL4* uniquely mark the PPL1 and PPM3 DAN clusters. Taken together, shorter flight bouts in *THD >Itpr$^{DN}$* flies and significant rescue of flight by *THD >Itpr$^+$* overexpression in flightless *itpr$^{ku}$* identifies an essential requirement for IP$_3$R function in the PPL1 and/or PPM3 DANs for flight bouts lasting upto ~300 s.

The two PPL1 clusters on each side of the brain consist of 12 pairs of neurons (*Mao, 2009*) and have been implicated in the maintenance of long flight bouts previously (*Pathak et al., 2015*). To further restrict flight modulating neuron/s in the PPL1 group we identified split*GAL4* strains that

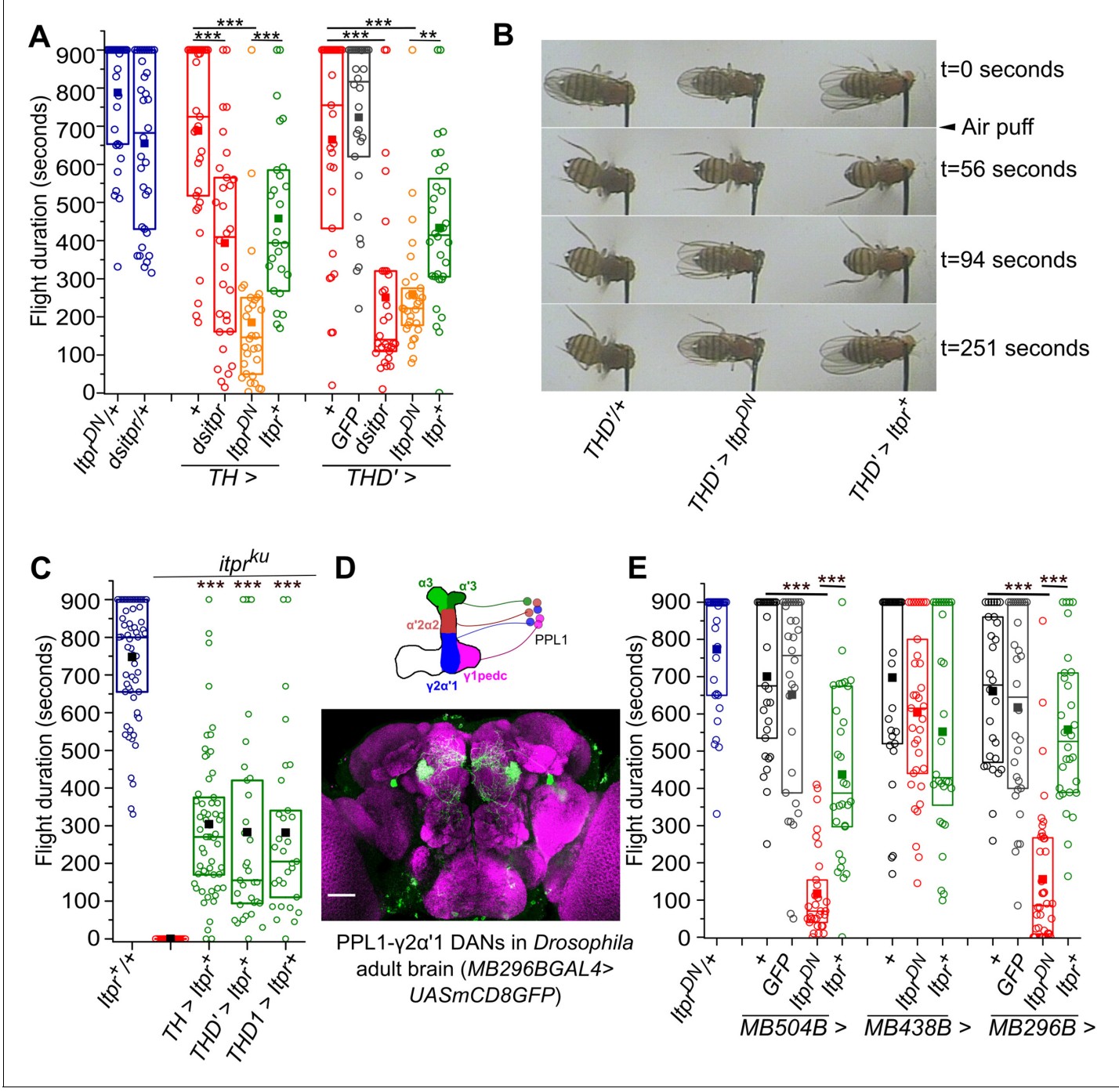

**Figure 2.** The IP$_3$R is required in central dopaminergic neurons for maintaining long flight bouts. (**A**) Flight deficits observed in flies expressing IP$_3$R RNAi (*dsitpr*), IP$_3$R$^{DN}$ (*Itpr$^{DN}$*) and IP$_3$R$^{WT}$ (*Itpr$^+$*) across all dopaminergic and TH-expressing cells (*TH*) as well as a subset of dopaminergic neurons (*THD'*). Flight times of flies of the indicated genotypes are represented as box plots where the box represents 25-75% of the distribution, each circle is the flight time of an individual fly, the small filled square represents mean flight time and the horizontal line is the median. Flight was tested in flies with GFP in *THD'* marked neurons as an over-expression control. (**B**) Snapshots from flight videos of air puff stimulated flight bouts in the indicated genotypes at the specified time points. (**C**) Box plots (as in A above) represent flight bout durations of control and *itpr* mutant flies (*itpr$^{ku}$*). Expression of a wild-type cDNA for the IP$_3$R (*UAS Itpr$^+$*) with indicated dopaminergic *GAL4s* rescued flight to a significant extent. (**D**) A schematic of PPL1 DANs projecting to different lobes of one half of the mushroom body (top); An adult brain with GFP expression (green) driven by *MB296BGAL4*. GFP expression is restricted to PPL1-γ2α'1 neurons and the γ2α'1 MB lobe. The brain neuropil is immunostained with anti Brp (purple). Scale bar = 50 μm. (**E**) Box plots (as in A) of flight bout durations in flies expressing IP$_3$R$^{DN}$ (*Itpr$^{DN}$*) and IP$_3$R$^{WT}$ (*Itpr$^+$*) in the indicated *PPL1 DAN* splitGAL4 strains, not

*Figure 2 continued on next page*

*Figure 2 continued*

significant at p < 0.05 by Mann-Whitney U test (for C) or Kruskal-Wallis test (for A and E). Comparisons for significance were with the control values except where marked by a horizontal line. Comparisons for significance were with *itpr* mutants in C.

The online version of this article includes the following video, source data, and figure supplement(s) for figure 2:

**Source data 1.** Flight duration for flies upon perturbing IP$_3$R signaling in dopaminergic neurons.
**Source data 2.** Flight duration of itpr mutant flies after overexpressing a wild type itpr transgene in various dopaminergic subs.
**Source data 3.** Flight duration of flies upon perturbing IP$_3$R signaling in PPL1 neurons.
**Source data 4.** Flight duration of flies upon perturbing IP$_3$R signaling pan neuronally.
**Source data 5.** Flight duration of *itpr* mutant flies after overexpressing a wild type *itpr* transgene in various neuronal subsets in a short flight assay.
**Source data 6.** Flight duration of flies upon perturbing IP$_3$R signaling in subsets of PPL1 .neurons.
**Figure supplement 1.** Perturbation of IP$_3$R signaling in neurons affects the duration of flight bouts.
**Figure 2—video 1.** Flight defect in *THD'> Itpr$^{DN}$* (Right) as compared to control (*THD'/+*, Left).
https://elifesciences.org/articles/62297#fig2video1

mark fewer PPL1 neuron/s and project to individual lobes of the Mushroom Body (MB; *Figure 2— figure supplement 1C* and *Figure 2D*; *Aso et al., 2014a*; *Aso and Rubin, 2016*). Amongst the identified split*GAL4* strains, flight deficits were observed by expression of IP$_3$R$^{DN}$ in PPL1 DANs projecting to the MB lobes α'2α2, α3, γ1peduncle and γ2α'1 (*MB504BGAL4*), but not with PPL1 DANs projecting to the MB lobes α'2α2, α3, and Y1peduncle (*MB438BGAL4*; *Figure 2E*, *Figure 2—source data 3*), suggesting PPL1-γ2α'1DANs as the primary focus of IP$_3$R function. Indeed, expression of *Itpr$^{DN}$* in PPL1-γ2α'1 DANs, marked by *MB296BGAL4*, resulted in significantly shorter flight bouts, when compared to *Itpr$^+$* expression in the same cells (*Figure 2E*). Expression of *Itpr$^{DN}$* in other PPL1 neurons failed to exhibit significant flight deficits (*Figure 2—figure supplement 1D*, *Figure 2— source data 6*). Thus *PPL1-γ2α'1* DANs require IP$_3$R function to sustain longer flight bouts. These data do not exclude a role for the IP$_3$R in PPM3 DANs that are also marked by *THD'GAL4* in the context of flight.

## The IP$_3$R is required in a late developmental window for adult flight

Shorter flight bouts in adults might arise either due to loss or change in properties of identified neurons during development. Alternately, the IP$_3$R might acutely affect the function of these neurons during flight in adults. To distinguish between these possibilities, we employed the TARGET system (*McGuire et al., 2003*) for temporal control of IP$_3$R$^{DN}$ expression in *PPL1* and *PPM3* DANs. TARGET uses a ubiquitously expressed temperature-sensitive repressor of GAL4 (Tub-GAL80$^{ts}$) that allows GAL4 expression at 29°, but not at lower temperatures. Hence expression of a GAL4 driven *UAS* transgene can be controlled by changing incubation temperatures from 18° to 29°. *THD'GAL4* driven *Itpr$^{DN}$* expression was restricted to the larval stages (data not shown), 0–48 hr after puparium formation (APF), 48–96 hr APF and for 0–3 days in adults. Flight deficits were observed upon expression of IP$_3$R$^{DN}$ during pupal development and not when expressed in adults. In pupae the deficit was most prominent by expression of the *Itpr$^{DN}$* during the shorter window of 48–96 hr APF (*Figure 3A*, *Figure 3—source data 1*).

Requirement for the IP$_3$R during the 48–96 hr APF window was independently confirmed by measuring flight after temporal expression of *Itpr$^+$* in dopaminergic neurons of *itpr$^{ku}$* (*Figure 3B*, *Figure 3—source data 2*). Flight in *itpr$^{ku}$* mutant animals was rescued to a significant extent by *Itpr$^+$* expression from 48 to 96 hr APF (277 ± 60.3 s) but not when expressed before and after. Interestingly, post-48 hr APF is also the time period during which levels of the IP$_3$R are upregulated in the pupal brain (*Figure 3C*, *Figure 3—source data 3*). Thus, expression from the *Itpr$^{DN}$* encoding transgene during the 48–96 hr interval presumably results in the formation of a majority of inactive IP$_3$R tetramers due to the presence of at least one IP$_3$R$^{DN}$ monomer. The cellular and physiological consequences of inactive IP$_3$Rs in pupal, and subsequently adult brains, was investigated next.

It is known that specification of central dopaminergic neurons is complete in late larval brains (*Hartenstein et al., 2017*) The number of PPL1 DANs was no different between controls and in flies expressing *Itpr$^{DN}$* in either *TH-D* or *MB296B* marked neurons (*Figure 3—figure supplement 1A*). This finding agrees with our observation that larval expression of *Itpr$^{DN}$* had no effect on adult flight. Moreover, projections from *MB296B* to the γ2α'1 MB lobes also appeared unchanged upon expression of *Itpr$^{DN}$* (*Figure 3—figure supplement 1B*).

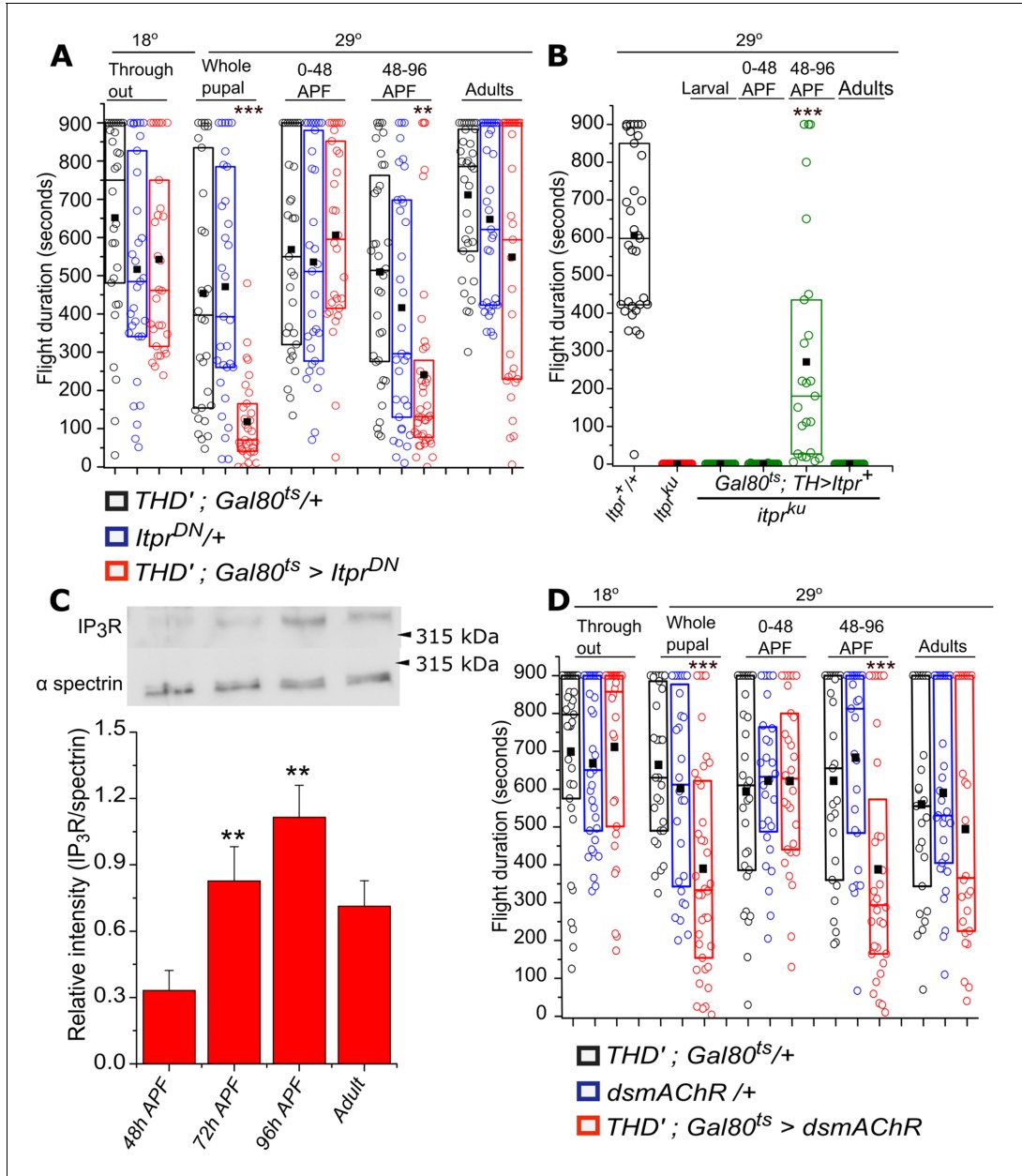

**Figure 3.** Adult flight phenotypes arise from late pupal expression of the IP3R and mAChR. (A) Box plots of flight bout durations in adults after temporal expression of IP3R$^{DN}$ in *THD'* neurons (*THD'; TubGAL80$^{ts}$ > Itpr$^{DN}$*) by transferring them to 29°C at the indicated stages of development. For adult expression, flies were transferred to 29°C immediately after eclosion and tested for flight after 3 days. (B) Box plots of flight bout durations after expressing the IP3R (*Itpr$^+$*) in dopaminergic cells of *itpr* mutants at the indicated stages of development. For stage specific expression a *TH; TubGAL80$^{ts}$* strain was used and the progeny transferred to 29°C at appropriate time developmental points. (C) Levels of the IP3R increase in late pupae between 72-96 hrs APF and plateau between 96hrs APF and adults (3 days). A representative western blot from lysates of dissected central nervous systems of *Canton S* probed with anti-IP3R and anti-spectrin is shown (top). Three independent lysates and blots were quantified (below). **$p < 0.05$, t-test. (D) Box plots with flight bout durations of adult flies after knockdown of the Muscarinic acetylcholine receptor (mAChR) in *THD'* neurons (*THD'; TubGAL80$^{ts}$ > dsmAChR*) at the indicated stages of pupal development and in adults. Box plot symbols are as described in methods, $n \geq 30$, ***$p < 0.005$ *$p < 0.05$, n.s., not significant by Kruskal-Wallis test (for A and D) and $n \geq 23$, ***$p < 0.005$, at $p < 0.05$ by Mann-Whitney U test (for B). All comparisons for significance were with the control values for A and C, with *itpr* mutants for B.

The online version of this article includes the following source data and figure supplement(s) for figure 3:

**Source data 1.** Flight duration of flies upon overexpression of IP3R$^{DN}$ at different stages of development.
**Source data 2.** Flight duration of *itpr* mutant flies after overexpressing a wild type *itpr* transgene at different stages of development.
**Source data 3.** Western blots of IP3R.
**Source data 4.** Flight duration of flies upon knockdown of mAChR in dopaminergic neurons at different stages of development.
*Figure 3 continued on next page*

*Figure 3 continued*
**Source data 5.** Flight duration of flies upon knockdown of mAChR in PPL1-γ2α′1 neurons.
**Figure supplement 1.** Expression of IP₃R$^{DN}$ and mAChR RNAi in *THD′* and *MB296B* marked neurons.

To understand the nature of signaling through the IP$_3$R, required during pupal development, we tested flight after knockdown of an IP$_3$/Ca$^{2+}$ linked GPCR, the muscarinic acetylcholine receptor (mAChR). From a previous study, it is known that neuronal expression of the mAChR during pupal development is required for adult flight (*Agrawal et al., 2013*). Adult flight bouts were significantly shorter upon stage specific knock-down (48–96 hr APF) of the mAChR in TH-D′ neurons with an RNAi under temporal control of the TARGET system (*Figure 3D*, *Figure 3—source data 4*). Further knockdown of mAChR in PPL1- *γ2α′1* DANs also manifested mild flight defects (*Figure 3—figure supplement 1C*, *Figure 3—source data 5*). These data suggest that the mAChR and IP$_3$-mediated Ca$^{2+}$ release are required during pupal maturation of a central brain circuit that functions for the maintenance of long flight bouts. Alternately, the pupal requirement may arise from the fact that both the IP$_3$R (*Figure 3A,B*) and the mAChR (*Figure 3D*) are synthesized in late pupal neurons, carried over to adult neurons where they have a slow turnover, and function during acute flight. Taken together our data support the idea that acetylcholine, a neurotransmitter, activates the mAChR on PPL1 DANs of late pupae and/or adults to stimulate Ca$^{2+}$ release through the IP$_3$R. Cellular changes arising from loss of IP$_3$ mediated Ca$^{2+}$ release were investigated next.

## Synaptic vesicle release and IP$_3$R are both required for the function of *PPL1-γ2α′1* DANs

The 48–96 hr time window of pupal development is when adult neural circuits begin to mature with the formation of synapses, some of which are eliminated whilst others are strengthened (*Akin et al., 2019*; *Consoulas et al., 2002*; *Zhang et al., 2016*). Synapse strengthening occurs when pre-synaptic neurotransmitter release leads to post-synaptic excitation/inhibition (*Andreae and Burrone, 2018*; *Baines, 2003*; *Baines et al., 2001*; *Pang et al., 2010*). We hypothesized that IP$_3$-mediated Ca$^{2+}$ release might modulate synaptic activity and hence lead to strengthening of synapses between *THD′* DANs and their post-synaptic partners during pupal development. To test this idea, the requirement for synaptic vesicle recycling in *THD′* and *PPL1-γ2α′1* DANs was investigated through pupal development. A transgene encoding a temperature-sensitive mutant of Dynamin, *Shibire*$^{ts}$ (*Kitamoto, 2001*), that prevents synaptic vesicle recycling at 29°C and hence blocks neurotransmitter release, was expressed during pupal development, by transfer to 29°C at the appropriate time interval. Loss of synaptic vesicle recycling in pupae resulted in significantly shorter flight bout durations of 419.9 ± 50.2 s (*THD′*) and 411.5 ± 57 s (*MB296B*) as compared to controls (*Figure 4A*, *Figure 4—source data 1* and *Figure 4—figure supplement 1A*, *Figure 4—source data 4*). The requirement for synaptic vesicle release was further restricted to 48–96 hr APF consistent with the requirement of IP$_3$R at the same time interval (*Figure 3A,B*). These data suggest that *THD′* marked dopaminergic neurons require both synaptic vesicle recycling and IP$_3$-mediated Ca$^{2+}$ release during late pupal development for their adult function.

The acute requirement for synaptic vesicle recycling in adult PPL1 (*THD′*) and *PPL1-γ2α′1* subset (*MB296B*) of DANs for maintenance of flight bouts was tested next. A previous report shows that synaptic vesicle recycling is required in PPL1 DANs marked by *THD1GAL4* during active flight (*Ravi et al., 2018*). We tested if synaptic vesicle release is also required in *THD′* and *MB296B* marked PPL1 DANs in adults. Importantly, flight bout durations were reduced significantly after acute inactivation (5 min) of synaptic vesicle recycling in PPL1 DANs (*THD′*) and the pair of *PPL1-γ2α′1* subset (*MB296B*) DANs in adults, supporting a requirement for neurotransmitter release from these neurons during flight (*Figure 4A*).

Impaired synaptic vesicle release from adult *PPL1-γ2α′1 DANs,* by expression of Shibire$^{ts}$, also reduces the ability of starved flies to identify a food source rapidly (*Tsao et al., 2018*). To understand if the IP$_3$R affects the function of *PPL1-γ2α′1* DANs in more than one behavioural context, we tested food-seeking behaviour of starved flies expressing *Itpr*$^{DN}$ in PPL1 (*THD′*) and *PPL1-γ2α′1* subset (*MB296B*) *DANs.* The food seeking index of hungry flies with IP$_3$R$^{DN}$ was reduced significantly as compared to controls (*Figure 4B and C*, *Figure 4—source data 2*, *Figure 4—video 1* and

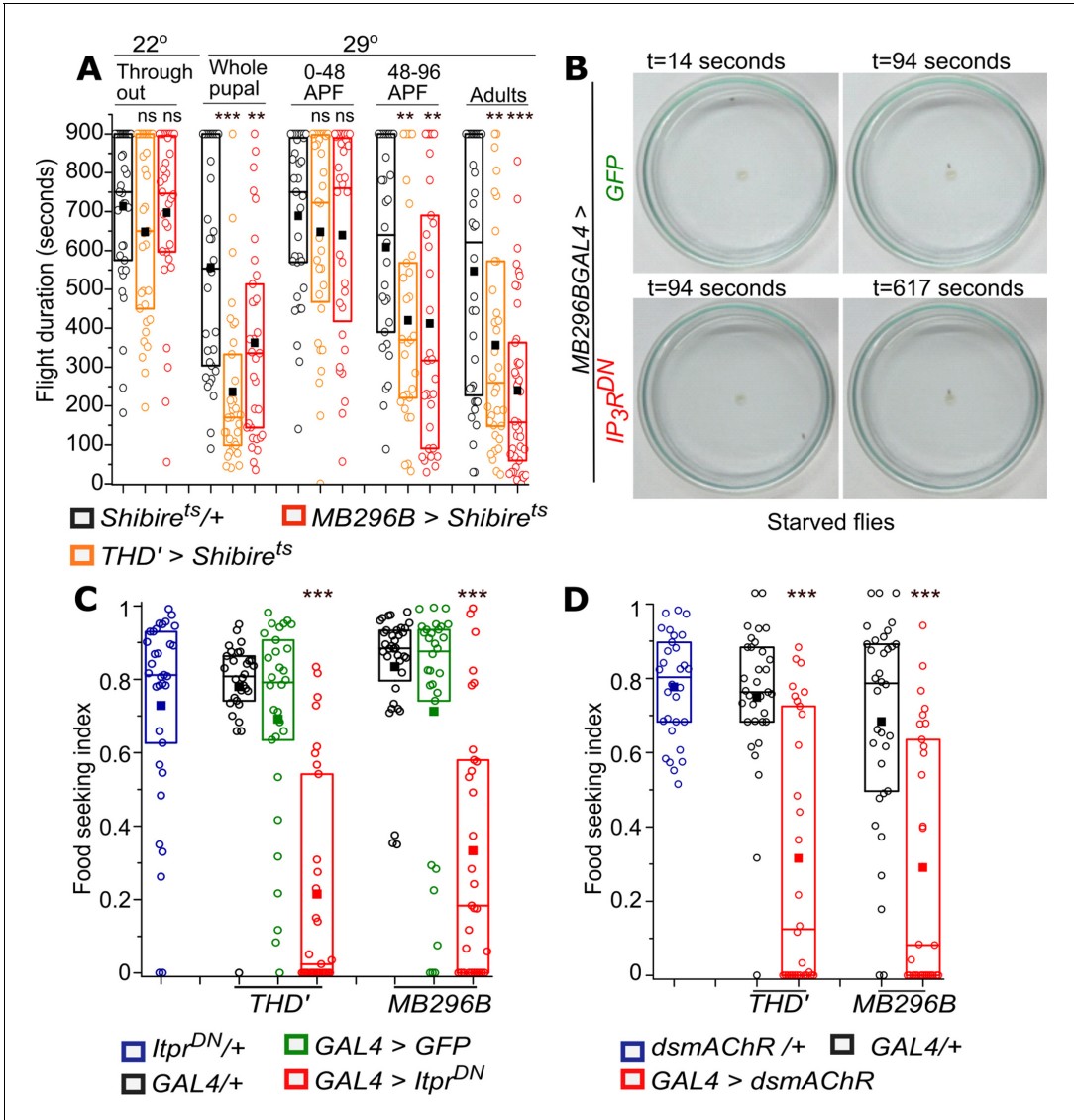

**Figure 4.** Synaptic vesicle recycling and IP$_3$R function are required in PPL1 dopaminergic neurons for adult flight and feeding. (**A**) Quantification of flight deficits observed upon blocking synaptic vesicle recycling by expression of a temperature-sensitive Dynamin mutant, *Shibire*$^{ts}$ in *THD'* and *MB296B* cells. n≥30, ***p< 0.005, **p< 0.05 by Mann Whitney U test. (**B**) Food seeking by hungry flies is significantly diminished upon expression of IP$_3$R$^{DN}$ in PPL1- γ2α′1 DANs (*MB296B*). Snapshots at the indicated time points from videos of starved male flies of the indicated genotype seeking a drop of yeast placed in the centre of a petriplate. (**C**) Quantification of food seeking behaviour in starved males of the indicated genotypes. Expression of IP$_3$R$^{DN}$ in *THD'* and *MB296B* cells reduced food seeking behaviour to a significant extent (red). Expression of *GFP* did not affect the behaviour (green). n≥30, ***p< 0.005 by Kruskal-Wallis test. (**D**) Quantification of food-seeking behaviour in starved males of the indicated genotypes. Knockdown of mAChR in *THD'* and *MB296B* cells reduced food-seeking behaviour to a significant extent (red), n≥30, ***p< 0.005 at by Kruskal-Wallis test.

The online version of this article includes the following video, source data, and figure supplement(s) for figure 4:

**Source data 1.** Flight duration of flies upon expression of *Shibire*$^{ts}$ in dopaminergic neurons at different developmental stages.
**Source data 2.** Food seeking index of starved male flies after overexpression of IP$_3$R$^{DN}$ in dopaminergic neurons.
**Source data 3.** Food seeking index of starved male flies upon knockdown of mAChR in dopaminergic neurons.
**Source data 4.** Flight duration of flies upon expression of *Shibire*$^{ts}$ in dopaminergic neurons, grown at 25˚C.
**Source data 5.** Food seeking index of fed male flies after overexpression of IP$_3$R$^{DN}$ in dopaminergic neurons.
**Source data 6.** Food seeking index of fed male flies upon knockdown of mAChR in dopaminergic neurons.
**Figure supplement 1.** Flight is unaffected and food seeking behaviour is normal in control genotypes and conditions.
**Figure 4—video 1.** Food-seeking behaviour in a control fly (*MB296BGAL4/+*).
https://elifesciences.org/articles/62297#fig4video1
**Figure 4—video 2.** Food-seeking behaviour upon expression of IP$_3$R$^{DN}$ in *MB296B* cells.
https://elifesciences.org/articles/62297#fig4video2

*Figure 4—video 2*). Food-seeking behaviour of fed flies of all genotypes tested appeared similar (*Figure 4—figure supplement 1B*, *Figure 4—source data 5*).

Next, we tested if knockdown of mAChR, the identified flight regulating GPCR (*Figure 3* and *Figure 3—figure supplement 1C*) that couples to IP$_3$/Ca$^{2+}$ signaling, also affected food seeking in starved flies. Knockdown of mAChR in either PPL1 (*THD'*) or *PPL1-γ2α'1* decreased the ability of starved males to find a yeast drop (*Figure 4D*, *Figure 4—source data 3*), while fed flies were similar to genetic controls (*Figure 4—figure supplement 1C*, *Figure 4—source data 6*). Thus cholinergic stimulation of IP$_3$/Ca$^{2+}$ in *PPL1-γ2α'1* DANs reduces the motivation for longer flight bouts as well as the motivation to search for food.

Taken together these data demonstrate a requirement for both synaptic vesicle recycling and IP$_3$-mediated Ca$^{2+}$ release in the modulation of behaviour by *PPL1-γ2α'1* DANs. Furthermore, they suggest that the IP$_3$R might regulate synaptic function in *PPL1-γ2α'1* DANs.

## The IP$_3$R affects neurotransmitter release from adult dopaminergic neurons

Decreased synaptic activity upon IP$_3$R$^{DN}$ expression might be a consequence of a reduction in synapse number during circuit maturation in pupae. Therefore, a pre-synaptic marker, synaptotagmin tagged to GFP (*syt.eGFP*; *Zhang et al., 2002*), that localises to synaptic vesicles was expressed in *THD'* marked neurons in the absence and presence of IP$_3$R$^{DN}$. Fluorescence of Syt.eGFP in the MB lobes was no different between control brains and in presence of the IP$_3$R$^{DN}$ (*Figure 5A*), suggesting that synapse numbers were unchanged by expression of IP$_3$R$^{DN}$. However, Syt.eGFP does not necessarily measure functional MB synapses. Acetylcholine and mAChR driven IP$_3$R – Ca$^{2+}$ signals in dopaminergic neurons might thus effect formation of functional MB synapses during circuit maturation in pupae. Loss of flight deficits observed upon inactivating synaptic vesicle recycling in late pupae supports this idea (*Figure 4A*).

Flight deficits also occur upon acute inactivation of synaptic vesicle recycling in adults (*Figure 4A*). Therefore, neurotransmitter release from *PPL1-γ2α'1* (*MB296B*) DANs in response to stimulation of the IP$_3$R was investigated next in adults. Presence of the mAChR was confirmed on *PPL1-γ2α'1* DANs in adults by measuring Ca$^{2+}$ release upon stimulation with the mAChR agonist Carbachol. Changes in GCaMP6m fluorescence in *MB296B* marked neurons post-Carbachol stimulation are shown in *Figure 5—figure supplement 1A*. The ability of Carbachol to stimulate dopamine release from *MB296B* marked neurons in the γ2α'1 MB-lobe was measured next. For this purpose, a recently designed fluorescent sensor for dopamine, GRAB$_{DA}$ (G-protein-coupled receptor-activation based DA sensor; *Sun et al., 2018*) was expressed in *MB296B* neurons followed by stimulation with Carbachol. GRAB$_{DA}$ consists of a dopamine receptor linked to cpEGFP such that its fluorescence increases upon binding of dopamine. Thus, a qualitative measure of dopamine release at the synaptic cleft is the change in GRAB$_{DA}$ fluorescence when recorded in the γ2α'1 MB lobe, the site of *PPL1-γ2α'1* synapses (*Figure 5B and C* and *Figure 5—source data 1*). Control brains, with detectable changes in GRAB$_{DA}$ fluorescence (ΔF/F) above an arbitrary value of 0.05 within 140 s of Carbachol stimulation, were classified as responders. Responding brains decreased from 77% (10/13 brains) in controls to 47% (8/17 brains) upon expression of IP$_3$R$^{DN}$. Moreover, GRAB$_{DA}$ fluorescence arising from dopamine release was significantly muted amongst responders expressing the IP$_3$R$^{DN}$ as compared to responders from control brains (*Figure 5C* and *Figure 5—figure supplement 1B*). Thus, expression of IP$_3$R$^{DN}$ in *MB296B* DANs reduced the synaptic release of dopamine at the γ2α'1 lobe. Interestingly, increase in GRAB$_{DA}$ fluorescence at the synaptic terminals in the MB was faster (*Figure 5C*) than the carbachol-stimulated change in GCaMP fluorescence measured in the cell body as measured by time taken to reach the peak response (*Figure 5—figure supplement 1A*). Perhaps, there exist a greater number of mAChRs near the synapse than on the soma of *MB296B* DANs. For example, mAChRs are also present on the processes of Kenyon cells in the mushroom body (*Bielopolski et al., 2019*; *Kondo et al., 2020*) and form a tripartite synapse with dendrites from MB output neurons (MB-ONs) and axonal processes of *MB296B* DANs. Although requirement for the IP$_3$R and the mAChR was restricted to 48–96 hr APF, these data suggest that the neurotransmitter acetylcholine stimulates dopamine release in adult DANs through mAChR-IP$_3$R signaling. The lack of flight deficits by adult-specific expression of IP$_3$R$^{DN}$ (*Figure 3A*) and knockdown of mAChR (*Figure 3D*) is probably due to perdurance of the respective proteins from late pupae through to adults. It remains possible that reduced strength of MB synapses, by pupal expression of IP$_3$R$^{DN}$ and

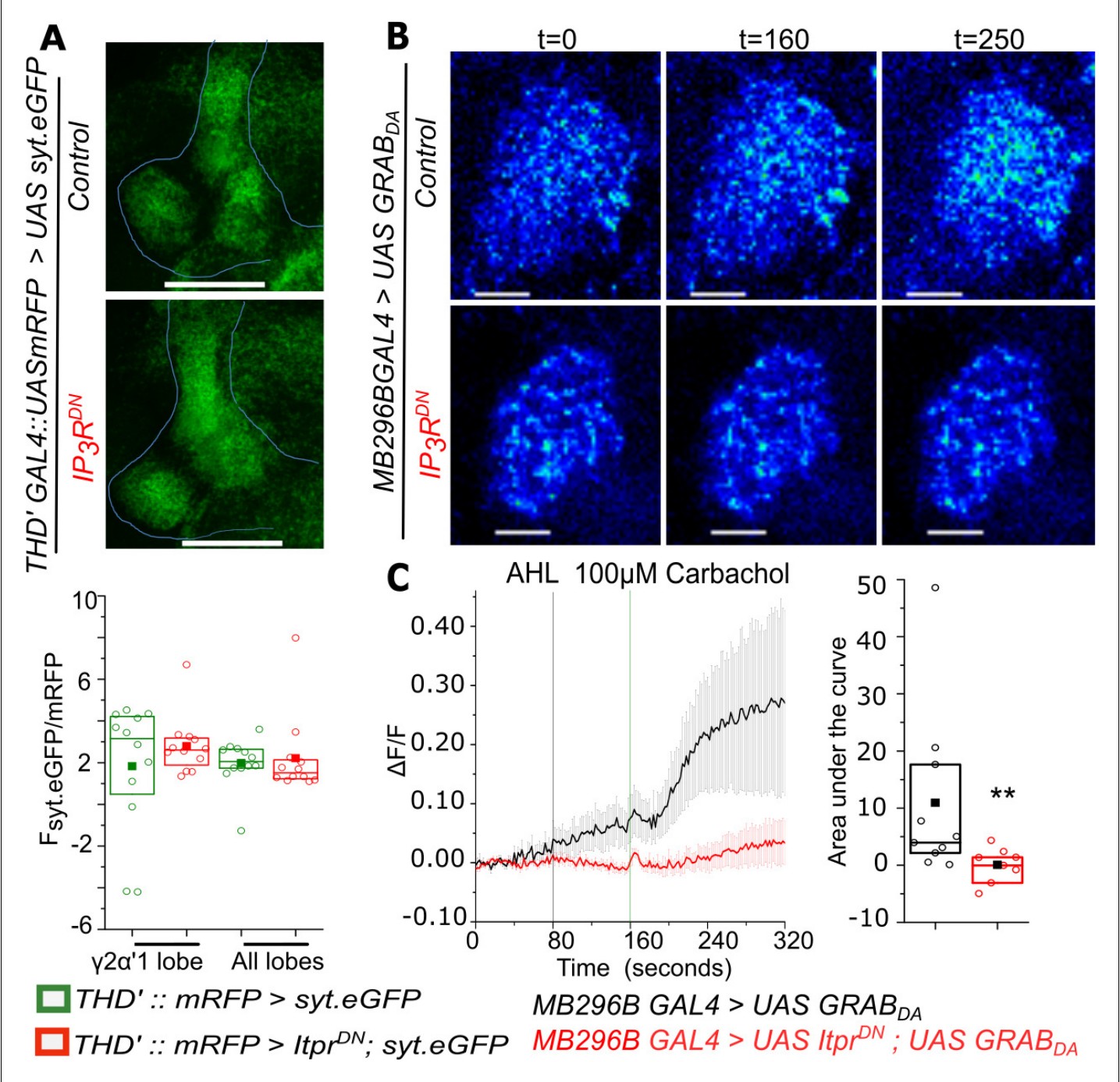

**Figure 5.** Carbachol-stimulated Dopamine release at the axonal terminals of *PPL1-γ2α′1* neurons is attenuated by expression of IP₃R$^{DN}$. (**A**) Representative images of a presynaptic marker synaptotagmin GFP (syt.eGFP) in lobes of the Mushroom body (top). Quantification of syt.eGFP fluorescence normalized to mRFP is shown below in the indicated MB regions marked by *THD′GAL4*. N = 6 brains, n.s. not significant at p < 0.05 by two tailed Mann-Whitney U test. Scale bars indicate 50 μm (**B**) Carbachol stimulated dopamine release at PPL1- γ2α′1 termini visualized by changes in GRAB$_{DA}$ fluorescence in a representative mushroom body lobe from brains of the indicated genotypes. Images were acquired at 2s per frame. Brighter fluorescence denotes increase in Dopamine. Scale bars = 10 μm (**C**) Average traces (± SEM) of normalized GRAB$_{DA}$ fluorescence responses (ΔF/F) from the right lobe of each brain upon addition of Carbachol (Green line) (panel on the left). Images were acquired at 2s per frame; Quantification of area under the curve was from the point of stimulation at 160s and up to 250 s. Box plots and symbols are as described in *Figure 2A*. *MB296B GAL4 > UAS GRAB$_{DA}$* N = 10 cells from 10 brains; *MB296B GAL4 > UAS Itpr$^{DN}$; UAS GRAB$_{DA}$* N = 8 cells from 8 brains. \*\*p< 0.05, Mann-Whitney U test.

The online version of this article includes the following source data and figure supplement(s) for figure 5:

*Figure 5 continued on next page*

*Figure 5 continued*

**Source data 1.** GRAB$_{DA}$ fluorescence (ΔF/F) traces from axonal termini of *MB296B*GAL4 marked DANs in individual brains (Black -*MB296B GAL4>UAS GRAB$_{DA}$*; Red-*MB296B GAL4>UAS Itpr$^{DN}$; UAS GRAB$_{DA}$*).

**Figure supplement 1.** Carbachol evoked Ca$^{2+}$ responses in the soma of PPL1- γ2α′1 DANs marked by *MB296B*GAL4.

concomitant inhibition of synaptic vesicle release during circuit maturation, also contribute to adult flight deficits.

## The IP$_3$R helps maintain membrane excitability of *PPL1-γ2α′1* neurons

In addition to the neuromodulatory input of acetylcholine, the *PPL1-γ2α′1* DANs are likely to receive direct excitatory inputs. Therefore, next we investigated if loss of IP$_3$/Ca$^{2+}$ signals alter the essential properties of neuronal excitability of *MB296B* neurons. Stimulation by activation of an optogenetic tool, *CsChrimson* (*Klapoetke et al., 2014*) was followed by measuring changes in fluorescence of the Ca$^{2+}$ sensor GCaMP in ex-vivo brain preparations. In neurons expressing IP$_3$R$^{DN}$ and GCaMP optogenetic stimulation of CsChrimsom did not evince a significant Ca$^{2+}$ response either during or after the red light stimulus, as compared with the control genotype expressing RFP and GCaMP (*Figure 6A,B* and *Figure 6—source data 1*). In agreement with recent findings, some *MB296B* neurons exhibit high basal activity even in the absence of optogenetic stimulation (*Figure 6—figure supplement 1*; *Siju et al., 2020*). Similar results were obtained by KCl - evoked depolarisation in the presence of 2 µM Tetrodotoxin (TTX) (*Figure 6—figure supplement 1A,B*, *Figure 6—source data 3*). TTX was added so as to prevent excitation by synaptic inputs from other neurons upon KCl addition. These data suggest that neurons with IP$_3$R$^{DN}$ fail to respond to a depolarising stimulus. This idea was tested directly by measuring the change in membrane potential in response to depolarisation, with a genetically encoded fluorescent voltage indicator, *Arclight* (*Cao et al., 2013*). There was an instant decline in Arclight fluorescence in control cells, while cells with IP$_3$R$^{DN}$ showed almost no change in fluorescence after KCl-mediated depolarisation (*Figure 6C,D*, *Figure 6—source data 2*). Taken together, these observations confirm that signaling through the IP$_3$R is also required for maintaining excitability of central neuromodulatory dopaminergic neurons.

## Discussion

An inducible IP$_3$R$^{DN}$ construct developed and used in this study allowed us to perform stage and cell-specific attenuation of IP$_3$R mediated calcium signaling in vivo. Consistent with well characterised phenotypes of IP$_3$R mutants (*Banerjee et al., 2004*), neuronal expression of IP$_3$R$^{DN}$ affected flight. Spatiotemporal studies identified a requirement for the IP$_3$R in a small subset of central dopaminergic neurons for maintenance of adult flight bouts as well as in the food-seeking behaviour of hungry flies. Inhibition of synaptic release in the identified dopaminergic subset also reduced the duration of flight bouts. Dopamine release at synapses in the MB γ2α′1 lobe was significantly attenuated in adults expressing the IP$_3$R$^{DN}$ (carried over from late pupae). These animals in addition exhibit reduced membrane excitability. Intracellular Ca$^{2+}$ signaling through the IP$_3$R is thus required to ensure optimal neuronal excitability and synaptic function in specific central dopaminergic neurons that appear to drive the motivation for both longer flight bouts and the search for food in a hungry fly (*Figure 7*).

## The IP$_3$R and synaptic release in pupae and adults

Temporal expression of IP$_3$R$^{DN}$ demonstrated a requirement during circuit maturation in late pupae but not in adults. Taken together with flight deficits that arise as a consequence of pupal inactivation of vesicle recycling (*Figure 4A*), these data suggest that Ca$^{2+}$ release through the IP$_3$R stimulates neurotransmitter release required for maturation of PPL1- γ2α′1 synapses with MBONs (*Aso et al., 2014b*; *Berry et al., 2018*) and possibly KCs (*Cervantes-Sandoval et al., 2017*) in late pupae. However, these data do not rule out an acute function for the IP$_3$R in adult dopaminergic neurons because functional wild-type tetramers of the IP$_3$R assembled during pupal development very likely mask the effect of adult specific expression of IP$_3$R$^{DN}$. The presence of perdurant IP$_3$R proteins in adults may be addressed in future by pupal expression of tagged proteins. Because short flight bouts are also observed upon acute inhibition of vesicle recycling in adults our data support a model

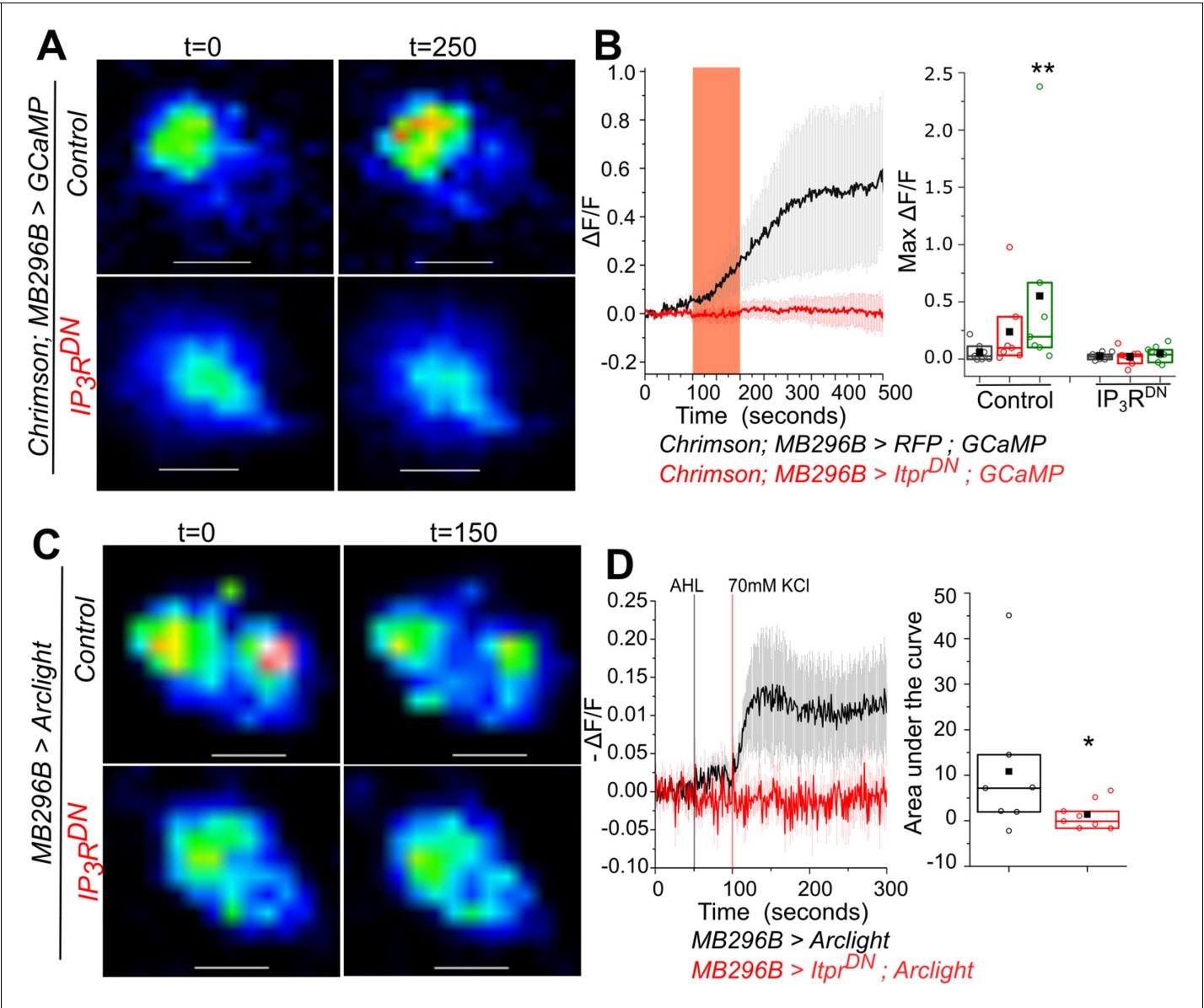

**Figure 6.** Optimal excitability of PPL1- γ2α′1 dopaminergic neurons requires the IP3R. (**A**) Optogenetic activation of PPL1- γ2α′1 DANs, with the red light activated Channelrhodopsin variant Chrimson, is attenuated by expression of IP3R$^{DN}$. Representative images of *MB296BGAL4* marked DANs of the indicated genotypes are shown with changes in GCaMP6m fluorescence before (t=0) and after (t=250s) a 100s pulse of red light (red bar). Images were acquired at 2s per frame. Warmer colors denote increase in [Ca$^{2+}$]. Scale bar = 5 μm. (**B**) Average traces (± SEM) of normalized changes in GCaMP6m fluorescence (ΔF/F) in *MB296BGAL4* marked DANs after activation by Chrimson (left); Quantification of areas under the curve are shown for before stimulation (0-100 sec, gray), during stimulation (100-200s, red) and after stimulation (200-500s, green); (right). Box plots and symbols are as described in *Figure 2A*. *MB296B GAL4 > UAS RFP; UAS GCaMP*, N = 7 cells from 7 brains; *MB296B GAL4 > UAS Itpr$^{DN}$; UAS GCaMP*, N = 8 cells from 8 brains **p< 0.05, Mann-Whitney U test. (**C**) Changes in membrane potential upon addition of KCl, visualized by expression of the voltage sensor Arclight. Representative images of Arclight responses in *MB296BGAL4* marked DANs are shown from the indicated genotypes and time points. Images were acquired at 1s per frame; Scale bar = 5 μm. Average traces (± SEM) of normalized changes in Arclight fluorescence (-ΔF/F) in *MB296BGAL4* marked DANs after addition of KCl (left). Images were acquired at 1s per frame; Quantification of area under the curve is from the point of stimulation at 100s up to 200s (right). Box plots and symbols are as described in *Figure 2A*. *MB296B GAL4 > UAS Arclight* N = 7 cells from 7 brains; *MB296B GAL4 > UAS Itpr$^{DN}$; UAS Arclight* N = 8 cells from 8 brains *p< 0.1, Mann-Whitney U test.

The online version of this article includes the following source data and figure supplement(s) for figure 6:

**Source data 1.** GCaMP6m fluorescence (ΔF/F) traces in*MB296BGAL4* marked DANs after activation by Chrimson in individual brains (Black -*MB296B GAL4>UAS RFP; UAS GCaMP*; Red-*MB296B GAL4>UAS Itpr$^{DN}$; UAS GCaMP*).

**Source data 2.** Arclight fluorescence (ΔF/F) traces in *MB296BGAL4* marked DANs after KCl induced depolarisation in individual brains (Black -*MB296B GAL4>UAS Arclight*; Red-*MB296B GAL4>UAS Itpr$^{DN}$; UAS Arclight*).

*Figure 6 continued on next page*

*Figure 6 continued*

**Source data 3.** GCaMP fluorescence (ΔF/F) traces in*MB296BGAL4*marked DANs after KCl induced depolarisation in individual brains (Black -*MB296B GAL4>UAS GCaMP*; Red-*MB296B GAL4>UASItpr^{DN}; UAS GCaMP*).

**Figure supplement 1.** Perturbation of IP$_3$R signaling in PPL1- γ2α′1 cells reduced KCl-evoked calcium response.

where mAChR stimulated ER-Ca$^{2+}$ signals through the IP$_3$R regulate synaptic release of dopamine from PPL1- γ2α′1 DANs during circuit maturation and during adult flight. Both direct and indirect effects of ER-Ca$^{2+}$ on synaptic vesicle release have been observed in vertebrates (*Gomez et al., 2020*; *Rossi et al., 2008*; *Sharma and Vijayaraghavan, 2003*) and *Drosophila* (*James et al., 2019*; *Klose et al., 2010*; *Richhariya et al., 2018*; *Shakiryanova et al., 2011*). Further studies are required to identify the molecular mechanism by which ER-Ca$^{2+}$ signals regulate dopamine release in PPL1- γ2α′1 DANs.

## Intracellular Ca$^{2+}$ signaling and neuronal excitability

Genetic manipulations that target intracellular calcium signaling are known to affect the intrinsic excitability of neurons. For example, Purkinje neurons in mice with cell-specific knockout of the ER-Ca$^{2+}$ sensor Stim1, that functions downstream of mGluR1, exhibit a decreased frequency of firing (*Ryu et al., 2017*). In *Drosophila* FMRFa Receptor mediated calcium signaling modulates the excitability of PPL1 dopaminergic neurons through CAMKII (*Ravi et al., 2018*). Reduced excitability of dopaminergic neurons expressing the IP$_3$R$^{DN}$ transgene (*Figure 6* and *Figure 6—figure supplement 1*) further supports a role for intracellular calcium signals in setting the threshold of membrane excitability in response to neuromodulatory signals, such as acetylcholine (this study), the FMRFa neuropeptide (*Ravi et al., 2018*), and glutamate (*Ryu et al., 2017*).

The cellular mechanism(s) by which IP$_3$/Ca$^{2+}$ signals regulate neuronal excitability probably vary among different classes of neurons. In addition to the direct activation of Ca$^{2+}$-dependant enzymes such as CamKII, previous reports have shown that knock-down of the IP$_3$R in *Drosophila* larval neurons alters their expression profile and specifically affects the expression of several membrane localised ion-channels (*Jayakumar et al., 2018*). Reduced translation of proteins due to IP$_3$R knockdown has also been demonstrated in peptidergic neurons (*Megha and Hasan, 2017*). Changes in membrane excitability could thus derive from direct regulation of ion channels by Ca$^{2+}$ and Ca$^{2+}$ dependant enzymes as well as an altered density of specific ion channels. Moreover, Ca$^{2+}$ release through the IP$_3$R regulates mitochondrial Ca$^{2+}$ entry (*Figure 1E*). Expression of IP$_3$R$^{DN}$ might thus impact neuronal firing and synaptic release by changes in cellular bioenergetics (*Cárdenas et al., 2010*; *Chouhan et al., 2012*) .

## PPL1- γ2α′1 DANs, flight and the search for food

Although insect flight is an innate behaviour the persistence of flight requires motivation presumably driven by neuromodulatory mechanisms and circuits in the central brain, that need to intersect with hard-wired flight circuits in the ventral ganglion. A role for the PPL1 cluster of dopaminergic neurons in modulating flight and longer flight bouts has been reported earlier (*Pathak et al., 2015*; *Ravi et al., 2018*). However, amongst the PPL1 cluster this is the first report identifying the two PPL1- γ2α′1 DANs as required for maintenance of flight. The PPL1- γ2α′1 DANs and their downstream Mushroom body output neuron (MBON- γ2α′1) also encode the state of hunger (*Tsao et al., 2018*), formation and consolidation of appetitive memory (*Berry et al., 2018*; *Felsenberg et al., 2017*; *Yamazaki et al., 2018*), fat storage (*Al-Anzi and Zinn, 2018*), and sleep (*Aso et al., 2014b*; *Sitaraman et al., 2015*). These studies identified the importance of modulated dopamine release as a motivational cue wherein dopamine release from PPL1- γ2α′1 DANs increased in starved flies (*Tsao et al., 2018*) and inhibited the activity of a cholinergic output neuron from the Mushroom Body (MBON- γ2α′1; *Tsao et al., 2018*). The MBON- γ2α′1 projects to another set of central DANs, the PAM neurons (*Felsenberg et al., 2017*), identified as flight promoting in a previous study (*Manjila et al., 2019*), as well as the fan shaped body (FSB) and the lateral accessory lobe (LAL; *Scaplen et al., 2020*). This leads to the hypothesis that inputs regarding the internal state reach the PPL1- γ2α′1 DANs through mAChR, amongst other GPCR (see below), driven IP$_3$/Ca$^{2+}$ signals to modulate dopamine release at the PPL1- γ2α′1 > MBON- γ2α′1 synapse, and the extent of

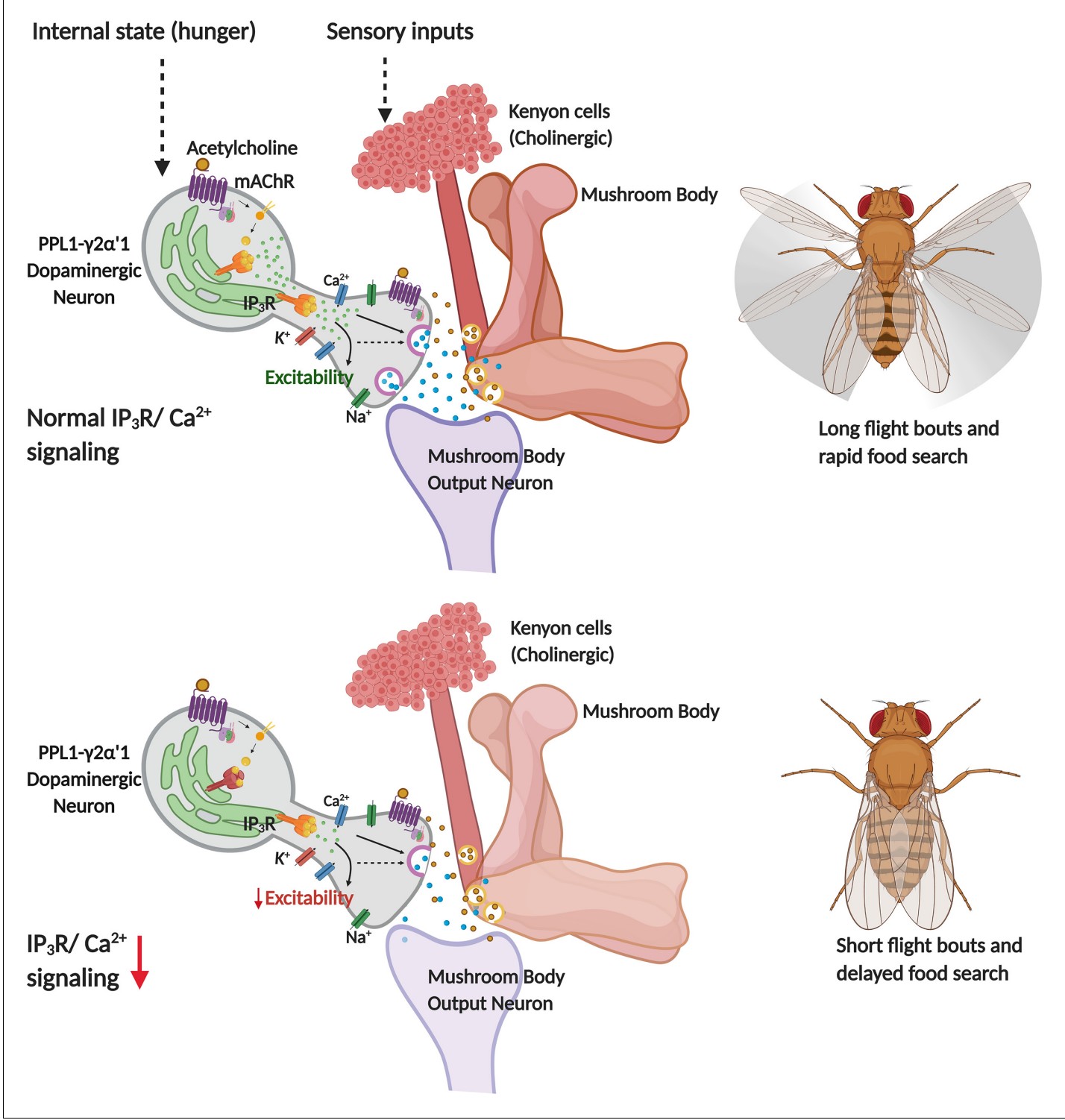

**Figure 7.** Schematic showing neuronal properties regulated by IP$_3$/Ca$^{2+}$ signals in central dopaminergic neurons for flight and food search behaviour in *Drosophila melanogaster*.

dopamine release changes the output strength of MBON - γ2α′1. In addition, continuous sensory inputs are essential for flight and these may reach PPL1- γ2α′1/MBON - γ2α′1 synapses through cholinergic Kenyon cells (*Cervantes-Sandoval et al., 2017*). In this context, carbachol-stimulated Dopamine release at the MB-γ2α′1 lobe may in part be post-synaptic to acetylcholine release from the

Kenyon cells. Altered MBON- γ2α′1 outputs might then regulate flight through their functional connections with the PAM-DANs, the LAL neurons and the FSB (*Scaplen et al., 2020*). Amongst these the LAL neurons signal to descending neurons (*Namiki et al., 2018*; *Namiki and Kanzaki, 2016*) that presumably connect with the flight circuit in the ventral ganglion. This idea is broadly supported by the observation that LAL neurons in locusts exhibit flight correlated activity changes (*Homberg, 1994*). The FSB processes visual inputs for flight navigation (*Weir and Dickinson, 2015*) and functions during turning behaviour (*Shiozaki et al., 2020*). In cockroaches, activation of the FSB induces turns (*Guo and Ritzmann, 2013*).

Prolonged flight bouts could serve to identify new sources of food for a hungry fly but at the same time they are energy intensive and very likely require multisensory integration with the inner state for continuous motivation. A recent review (*Lee and Wu, 2020*) describes a framework of 'homeostatic motivation 'as an integral part of such motivated behaviour, consisting of neural circuits with a 'sensor, integrator and effector'. The PPL1- γ2α′1 DANs receive inputs for assessing the internal state through multiple GPCRs including certain neuropeptide receptors such as sNPFR, Insulin receptor and AstA receptor and a serotonin receptor 5HT1B (*Albin et al., 2015*; *Hergarden et al., 2012*; *Root et al., 2011*; *Tsao et al., 2018*). Each of these signals appear to integrate hunger with food seeking behaviour (*Tsao et al., 2018*). We propose that modulated activity in the PPL1- γ2α′1 DANs serves as a 'sensor' followed by inhibition of MBON - γ2α′1, that forms an important multisensory 'integrating component' for control of downstream 'effector' circuits. Appropriate MBON-γ2α′1 outputs are required to maintain long flight bouts, similar to the maintenance of wakefulness (*Aso et al., 2014b*; *Sitaraman et al., 2015*) and increase in the search for food when hungry (*Tsao et al., 2018*). The integration of external cues that stimulate flight with the internal states of hunger and wakefulness very likely serve important functions of survival in the wild.

# Materials and methods

## Key resources table

| Reagent type (species) or resource | Designation | Source or reference | Identifiers | Additional information |
|---|---|---|---|---|
| Genetic reagent (*D. melanogaster*) | nSybGAL4 | Bloomington *Drosophila* Stock Center | RRID:BDSC_51635 | |
| Genetic reagent (*D. melanogaster*) | UAS GCaMP6m | Bloomington *Drosophila* Stock Center | RRID:BDSC_42750 | |
| Genetic reagent (*D. melanogaster*) | UAS GCaMP6m | Bloomington *Drosophila* Stock Center | RRID:BDSC_42748 | |
| Genetic reagent (*D. melanogaster*) | UAS Arclight | Bloomington *Drosophila* Stock Center | RRID:BDSC_51056 | |
| Genetic reagent (*D. melanogaster*) | UAS mCD8GFP | Bloomington *Drosophila* Stock Center | RRID:BDSC_5130 | |
| Genetic reagent (*D. melanogaster*) | UAS Dicer | Bloomington *Drosophila* Stock Center | RRID:BDSC_24648 | |
| Genetic reagent (*D. melanogaster*) | UAS Chrimson | Bloomington *Drosophila* Stock Center | RRID:BDSC_55137 | |
| Genetic reagent (*D. melanogaster*) | UAS syt.eGFP | Bloomington *Drosophila* Stock Center | RRID:BDSC_6926 | |
| Genetic reagent (*D. melanogaster*) | UAS RFP | Bloomington *Drosophila* Stock Center | RRID:BDSC_32218 | |

*Continued on next page*

Continued

| Reagent type (species) or resource | Designation | Source or reference | Identifiers | Additional information |
|---|---|---|---|---|
| Genetic reagent (*D. melanogaster*) | MB058BGAL4 | Bloomington *Drosophila* Stock Center | RRID:BDSC_68278 | |
| Genetic reagent (*D. melanogaster*) | MB296BGAL4 | Bloomington *Drosophila* Stock Center | RRID:BDSC_63308 | |
| Genetic reagent (*D. melanogaster*) | MB304BGAL4 | Bloomington *Drosophila* Stock Center | RRID:BDSC_68367 | |
| Genetic reagent (*D. melanogaster*) | MB320CGAL4 | Bloomington *Drosophila* Stock Center | RRID:BDSC_68253 | |
| Genetic reagent (*D. melanogaster*) | MB630BGAL4 | Bloomington *Drosophila* Stock Center | RRID:BDSC_68334 | |
| Genetic reagent (*D. melanogaster*) | MB438BGAL4 | Bloomington *Drosophila* Stock Center | RRID:BDSC_68326 | |
| Genetic reagent (*D. melanogaster*) | MB504BGAL4 | Bloomington *Drosophila* Stock Center | RRID:BDSC_68329 | |
| Genetic reagent (*D. melanogaster*) | UAS mAChR RNAi (dsmAChR) | Vienna *Drosophila* Resource Center RRID:SCR_013805 | VDRC_101407 | |
| Genetic reagent (*D. melanogaster*) | UAS itprRNAi (dsitpr) | National Institute of Genetics | NIG_1063 R-2 | |
| Genetic reagent (*D. melanogaster*) | UAS Itpr$^+$ | *Venkatesh et al., 2001* | RRID:BBSC_30742 | |
| Genetic reagent (*D. melanogaster*) | THGAL4 | *Friggi-Grelin et al., 2003* (DOI:10.1002/neu.10185) | | Gift from Serge Birman (CNRS, ESPCI Paris Tech, France) |
| Genetic reagent (*D. melanogaster*) | THD' GAL4 | *Liu et al., 2012* (DOI:10.1016/j.cub.2012.09.008) | | Gift from Mark N Wu (Johns Hopkins University, Baltimore) |
| Genetic reagent (*D. melanogaster*) | THD1 GAL4 | *Liu et al., 2012* (DOI:10.1016/j.cub.2012.09.008) | | Gift from Mark N Wu (Johns Hopkins University, Baltimore) |
| Genetic reagent (*D. melanogaster*) | THC' GAL4 | *Liu et al., 2012* (DOI:10.1016/j.cub.2012.09.008) | | Gift from Mark N Wu (Johns Hopkins University, Baltimore) |
| Genetic reagent (*D. melanogaster*) | UAS GRAB$_{DA}$ | *Sun et al., 2018* (DOI:10.1016/j.cell.2018.06.042) | | Gift from Yulong Li (Peking University School of Life Sciences, Beijing, China) |
| Genetic reagent (*D. melanogaster*) | UAS Shibire$^{ts}$ | *Kitamoto, 2001* (DOI:10.1002/neu.1018) | | Gift from Toshihiro Kitamoto (University of Iowa, Carver College of Medicine, Iowa) |
| Genetic reagent (*D. melanogaster*) | vGlut$^{VGN6341}$GAL4 | *Syed et al., 2016* (DOI:10.7554/eLife.11572) | | Gift from K. Vijayraghavan (NCBS, India) |
| Genetic reagent (*D. melanogaster*) | UAS mitoGCaMP | *Lutas et al., 2012* (DOI:10.1534/g3.111.001586) | | Gift from Fumiko Kawasaki (Pennsylvania State University, Pennsylvania) |
| Genetic reagent (*D. melanogaster*) | UAS TubGAL80$^{ts}$ | *Pathak et al., 2015* (DOI:10.1523/JNEUROSCI.1680–15.2015) | | Generated by Albert Chiang, NCBS, Bangalore, India |

*Continued*

| Reagent type (species) or resource | Designation | Source or reference | Identifiers | Additional information |
|---|---|---|---|---|
| Genetic reagent (*D. melanogaster*) | UAS Itpr*DN* | This paper | | Transgenic *Drosophila* with a Dominant negative *Drosophila* IP$_3$R cDNA |
| Genetic reagent (*D. melanogaster*) | *itpr* gene mutant (*itpr*$^{ug3}$) | *Joshi et al., 2004* (DOI:10.1534/genetics.166.1.225) | RRID:BDSC_30738 | |
| Genetic reagent (*D. melanogaster*) | *itpr* gene mutant (*itpr*$^{ka1091}$) | *Joshi et al., 2004* (DOI:10.1534/genetics.166.1.225) | RRID:BDSC_30739 | |
| Antibody | Rabbit anti GFP (polyclonal) | Life Technologies, Thermo Fisher | RRID:AB_221570, Cat # A-6455 | IHC: 1:10,000 |
| Antibody | Mouse anti-bruchpilot (monoclonal) | *Wagh et al., 2006* (DOI:10.1016/j.neuron.2006.02.008) | | Gift from Eric Buchner, University of Wuerzburg, Germany (dilution 1:150) |
| Antibody | Anti-mouse Alexa Fluor 568 (polyclonal) | Life Technologies, ThermoFisher Scientific | Cat# A-11004, RRID:AB_2534072 | IHC: 1:400 |
| Antibody | Anti-rabbit Alexa Fluor 488 (polyclonal) | Life Technologies, ThermoFisher Scientific | Cat# A-11008, RRID:AB_143165 | IHC: 1:400 |
| Antibody | Rabbit anti-IP$_3$R (polyclonal) | *Agrawal et al., 2009* (DOI:10.1371/journal.pone.0006652) | IB-9075 | WB: 1:300; Gift from Ilya Bezprozvanny (UT South Western, USA). |
| Antibody | Mouse anti-spectrin (monoclonal) | Developmental Studies Hybridoma Bank | Cat# 3A9 (323 or M10-2), RRID:AB_528473 | WB: 1:50 |
| Antibody | Anti-mouse HRP (polyclonal) | Cell Signaling Technology | Cat# 7076, RRID:AB_330924 | WB: 1:3000 |
| Antibody | Anti-rabbit HRP (polyclonal) | ThermoFisher Scientific | Cat#32260, RRID:AB_1965959 | WB: 1:5000 |
| Chemical compound, drug | Low melt Agar | Invitrogen | Cat# 16520–050 | |
| Chemical compound, drug | Carbachol | Sigma Aldrich | Cat# C4382 | |
| Chemical compound, drug | All trans Retinal | Sigma Aldrich | Cat# R2500 | |
| Chemical compound, drug | EcoR1 | New England Biolabs | Cat# R0101S | Restriction enzyme |
| Chemical compound, drug | Xho1 | New England Biolabs | Cat# R0146S | Restriction enzyme |
| Chemical compound, drug | AatII | New England Biolabs | Cat# R0117S | Restriction enzyme |
| Chemical compound, drug | Eag1 | New England Biolabs | Cat# R0505S | Restriction enzyme |
| Strain, strain background (*Escherichia coli*) | Sure Competent cells | Stratagene, Agilent Technologies | Cat# 200238 | Maintained in the lab |
| Commercial assay or kit | Quick Ligation Kit | New England Biolabs | Cat# M2200S | |
| Commercial assay or kit | WesternBright ECL kit | Advansta | Cat# K-12045-D20 | |
| Commercial assay or kit | QuikChange II XL Site-Directed Mutagenesis Kit | Agilent | Cat# 200522 | |
| Software, algorithm | Origin 8 | Origin lab | RRID:SCR_014212 | |
| Software, algorithm | Fiji/ImageJ | National Institutes of Health | RRID:SCR_002285 | |

## Fly stocks

*Drosophila* strains used in this study were reared on cornmeal media, supplemented with yeast. Flies were maintained at 25 ℃, unless otherwise mentioned under 12:12 light: dark cycle. WT strain of *Drosophila* used was *Canton S*.

## Single flight assay

Flight assays were performed according to *Manjila and Hasan, 2018*. Briefly, 3–5 day old flies of either sex were tested in batches of 8–10 flies. They were anaesthesized on ice for 2–3 min and then tethered between their head and thorax using a thin metal wire and nail polish. Once recovered, mouth blown air puff was given as a stimulus to initiate flight and flight time was recorded for each fly till 15 min. For all control genotypes, *GAL4* or *UAS* strains were crossed to wild type strain, *Canton S*. Flight time data is represented in the form of boxplots using Origin software (OriginLab, Northampton, MA). Each box represents 25th to 75th percentile, each open circle represents flight duration of a single fly, solid squares represent the mean and the horizontal line in each box represents the median.

For Gal80$^{ts}$ experiments, larvae, pupae, or adults were maintained at 18 ℃ and transferred to 29℃ only at the stage when the *UAS* transgene needed to be expressed. Flight assay was done at 25℃. For adult specific expression, flies were grown at 18 ℃ and transferred to 29℃ immediately after eclosion for 2–3 days. For experiments involving *Shibire$^{ts}$*, larvae, pupae or adults were maintained at 22℃ and transferred to 29℃ only at the stage when the *Shibire$^{ts}$* needs to be activated. Flight assay was done at 25℃ except for adult-specific activation in which flies were shifted to 29℃ 10 min before flight assay and then maintained at 29℃ during the experiment.

For optogenetic experiments, flies were transferred to media containing 200 mM all-trans-retinal (ATR) and reared in dark for 2–3 days before imaging experiments.

## Ex vivo live imaging

Adult brains were dissected in Adult hemolymph-like (AHL) saline (108 mM NaCl, 5 mM KCl, 2 mM CaCl$_2$, 8.2 mM MgCl$_2$, 4 mM NaHCO$_3$, 1 mM NaH$_2$PO$_4$, 5 mM trehalose, 10 mM sucrose, 5 mM Tris, pH 7.5) while larval brains were dissected in HL3 (70 mm NaCl, 5 mm KCl, 20 mm MgCl$_2$, 10 mm NaHCO$_3$, 5 mm trehalose, 115 mm sucrose, 5 mm HEPES, 1.5 mm Ca$^{2+}$, pH 7.2). The dissected brain was mounted on culture dish with anterior side up for recording from cell while posterior side up for imaging mushroom body. They were then embedded in 6 µl of 1% low-melt agarose and bathed in AHL. Images were taken as a time series on an XY plane using a 20x objective on an Olympus FV3000 inverted confocal microscope (Olympus Corp.). Acquisition time is different for different experiments and is described in the figure legends. GCaMP6m, Arclight and GRAB$_{DA}$ signals were captured using the 488 nm excitation laser line while 633 nm laser was used for optogenetic stimulation of Chrimson.

Raw fluorescence data were extracted from the marked ROIs using a time series analyzer plugin in Fiji (Balaji, https://imagej.nih.gov/ij/plugins/time-series.html). ΔF/F was calculated using the following formula for each time point (t): ΔF/F = (F$_t$-F$_0$)/F$_0$, where F$_0$ is the average basal fluorescence of the first 20 frames. Out of the 2 or 3 cells visualised in the brain, the cell which responded the best was taken for further analysis in every case. For analysis of fluorescence changes with GRAB$_{DA}$ the right lobe was chosen arbitrarily. Responses for the left lobe are included in a source data and are similar to the right lobe.

To quantify response to stimuli, we calculated area under the curve (AUC). Area under the curve was calculated from the point of stimulation till mean peak response was reached using Microsoft Excel (Microsoft). Time frame for calculating AUC is mentioned in figure legends. AUC is represented as boxplots using Origin software (OriginLab, Northampton, MA). Each box represents 25th to 75th percentile, each open circle represents flight duration of a single fly, solid squares represent the mean and the horizontal line in each box represents the median.

## Generation of IP$_3$R$^{DN}$

Five *itpr* residues in *Drosophila itpr* cDNA (*Sinha and Hasan, 1999*) were mutated using site directed mutagenesis kit (Agilent). The oligonucleotide CAGAGATCGGCAG**C**AATTGCTGC**AG**GAA-CAGTACATCC was used to change K530/R533 to Q while GTACCACGTCTTTCTGC

AGACCACCGGACGCACCAG was used to change R272 to Q. All mutations were confirmed using Sanger's sequencing. Mutated *itpr* cDNA was subcloned in *UAS attB* vector (*Bischof et al., 2007*). *UAS Itpr^DN* plasmid was then microinjected in fly embryos at the NCBS fly facility to obtain stable fly strains using standard protocols of fly embryo injection.

### Food-seeking assay

Food-seeking assay was performed according to *Tsao et al., 2018*. Briefly, 18–20 hr starved males (4 days old) of specified genotypes were introduced in petridish (dimensions) with drop of yeast solution in the centre. The yeast solution was prepared by mixing 0.2 g of yeast with 1 g of sucrose in 5 ml distilled water and incubated in a 28°C shaking incubator (170 rpm) for 16 hr. Starved males were then allowed to search for food, and it was considered having found food if it rested for 3 s or longer on food drop. Food-seeking index was calculated as: [Total assay time (600 s) - the time taken to locate food (sec)]/Total assay time (600 s).

### Immunohistochemistry

Immunohistochemistry was performed on dissected adult brains as described in *Pathak et al., 2015*. Briefly, brains were dissected in 1x PBS, followed by fixation in 4% paraformaldehyde for 30 min at room temperature and then 3–4 washings with 0.2% phosphate buffer, pH 7.2 containing 0.2% Triton-X 100 (PTX). They were then blocked in 0.2% PTX containing 5% normal goat serum for four hours at 4°C and incubated overnight with primary antibodies. Next day, they were washed three to four times with 0.2% PTX at room temperature, and then incubated with the respective fluorescent secondary antibodies for 2 hr at room temperature. The primary antibodies used were: rabbit anti-GFP, mouse anti-bruchpilot (anti-brp) antibody. Fluorescent secondary antibodies used at were anti-mouse Alexa Fluor 568 and anti-rabbit Alexa Fluor 488. Confocal images were obtained on the Olympus Confocal FV3000 microscope (Olympus Corp.) with a 20x or with a 40x objective. Images were visualized using Fiji.

Mean intensity fluorescence was obtained for Syt.eGFP and mRFP by averaging intensity values between anterior and posterior limits of the structure.

### Western blots

Between 5 to 10 pupal and adult brains or adult heads of appropriate genotypes were dissected in cold PBS and were homogenized in 30 µl of homogenizing buffer (25 mM HEPES, pH 7.4, 150 mM NaCl, 5% glycerol, 1 mM DTT, 1% Triton X-100, and 1 mM PMSF). 15 µl of the homogenate was run on a 5% SDS-polyacrylamide gel. The protein was transferred to a nitrocellulose membrane by standard protocols. Membrane was then blocked with 5% skim milk followed by incubation with primary antibody at 4°C overnight. The affinity-purified anti-IP$_3$R rabbit polyclonal antibody (IB-9075) was used at a dilution of 1:300 and mouse anti-spectrin antibody was used as a loading control for IP$_3$R. Secondary antibodies used were anti-mouse HRP and anti-rabbit HRP. The protein was then detected on the blot by a chemiluminiscent detection solution. First the spectrin antibody was used to detect protein. The blot was then washed with 3% glacial acetic acid for 40 min and re-probed with IP$_3$R antibody.

### Statistical tests

Non-parametric tests were employed to test significance for data that did not follow a normal distribution. Significant differences between experimental genotypes and relevant controls was tested either with the Kruskal-Wallis test followed by Dunn's multiple comparison test (for multiple comparisons) or with Mann-Whitney U tests (for pairwise comparisons). Data with normal distribution were tested by the Student's T-test. All statistical tests were performed using Origin 8.0 software. Statistical tests and p-values are mentioned in each figure legend. Source file two is provided with information for all the statistical tests performed.

Model in *Figure 7* was created using Biorender (BioRender.com).

## Acknowledgements

This study was supported by grants from DST-SERB and NCBS-TIFR to GH AS was supported by a fellowship from the National Centre for Biological Sciences, TIFR. We thank the Fly Facility, Sequencing facility and Central Imaging and Flow Cytometry Facility at NCBS.

## Additional information

### Funding

| Funder | Grant reference number | Author |
|---|---|---|
| Department of Science and Technology | National fellowship | Gaiti Hasan |
| National Centre for Biological Sciences | Core Grant | Gaiti Hasan |
| National Centre for Biological Sciences | Graduate fellowship | Anamika Sharma |

The funders had no role in study design, data collection and interpretation, or the decision to submit the work for publication.

### Author contributions

Anamika Sharma, Conceptualization, Formal analysis, Validation, Visualization, Methodology, Writing - original draft; Gaiti Hasan, Conceptualization, Resources, Supervision, Funding acquisition, Project administration, Writing - review and editing

### Author ORCIDs

Anamika Sharma (iD) https://orcid.org/0000-0003-4063-4748
Gaiti Hasan (iD) https://orcid.org/0000-0001-7194-383X

### Decision letter and Author response

Decision letter https://doi.org/10.7554/eLife.62297.sa1
Author response https://doi.org/10.7554/eLife.62297.sa2

## Additional files

### Supplementary files

• Transparent reporting form

### Data availability

All data generated or analysed during this study are included in the manuscript and supporting files.

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
