## [Decision Letter]

**Acceptance summary:**

The authors report experiments on *Drosophila* to show that the proper function of an IP_3_ receptor in a small subset of dopaminergic neurons is required for flight behavior and food search. Technically, the authors report a novel dominant-negative mutant for of the IP_3_ receptor to interfere with its function. Physiologically, the IP_3_ receptor-dependent impairment in the function of the dopaminergic neurons affects both synaptic vesicle release and excitability, Also, muscarinic acetylcholine receptors are required for proper development of the flight-modulating circuit.

The role of dopamine in the brain of *Drosophila* (as a model for general dopamine and brain function) is at the center of current research and is studied by a large number of laboratories. More and more types of behavior are discovered that are modulated by dopaminergic neurons, and particularly those innervating the mushroom body. Therefore, the study is of very high interest for researchers working on *Drosophila*, but also to a broader readership.

**Decision letter after peer review:**

Thank you for submitting your article "Modulation of flight and feeding behaviours requires presynaptic IP_3_Rs in dopaminergic neurons" for consideration by *eLife*. Your article has been reviewed by three peer reviewers, and the evaluation has been overseen by Ronald Calabrese as the Senior Editor and Reviewing Editor. The reviewers have opted to remain anonymous.

The reviewers have discussed the reviews with one another and the Reviewing Editor has drafted this decision to help you prepare a revised submission.

Summary:

The authors report experiments on *Drosophila* to show that the proper function of an IP_3_ receptor in a small subset of dopaminergic neurons is required for flight behavior and food search. Technically, the authors report a novel dominant-negative mutant for of the IP_3_ receptor to interfere with its function. Physiologically, the IP_3_ receptor-dependent impairment in the function of the dopaminergic neurons affects both synaptic vesicle release and excitability, Also, muscarinic acetylcholine receptors are required for proper development of the flight-modulating circuit.

The role of dopamine in the brain of *Drosophila* (as a model for general dopamine and brain function) is at the center of current research and is studied by a large number of laboratories. More and more types of behavior are discovered that are modulated by dopaminergic neurons, and particularly those innervating the mushroom body. Therefore, the study is of very high interest for researchers working on *Drosophila*, but also to a broader readership.

Essential revisions:

While there was considerable enthusiasm for the paper, there were concerns. These are expressed in detail in the appended reviews. Three particularly important concerns emerge from this reviews and subsequent discussion among the reviewers.

1) There were concerns about the normality tests and reanalysis to avoid pseudo-replication that must be addressed.

2) The Discussion should be made clearer and expanded to encompass more of the literature. Specifically, the authors should expand upon the final section of the Discussion to discuss more about 1) the potential context for cholinergic modulation of the PPL1-γ2α'1 DANs (For example, consider where the acetylcholine signal onto DANs might come from. DANs may not be entirely presynaptic to Kenyon cells but might also receive input from Kenyon cells.), 2) the proposed role of these DANs (which have been studied in several contexts) and 3) modulation of innate behavior in general. The paper begins with the importance of modulating innate behavior, but the discussion on this topic is spare and focused almost entirely on research on the mushroom bodies of *Drosophila*. The Discussion section leans heavily on summarizing the results, rather than making connections to work in other systems or networks.

3) One common point raised by all reviewers was the need for expression of the *itpr^DN^* during pupation which could have been due to either the perdurance of endogenous *itpr* vs. a developmental effect caused by the *itpr^DN^* (the authors fully acknowledge the issue). This section raised many questions that aren't within the scope of this study, nor are easily resolved. Nevertheless, the authors must expand upon the implications of these results and suggest future studies will needed to resolve the issue.

Reviewer #1:

The authors report experiments on *Drosophila* to show that the proper function of an IP_3_ receptor in a small subset of dopaminergic neurons is required for flight behavior. Most interesting is the fact that the requirement is restricted to a time point during pupal development. Technically, the authors report a novel dominant-negative mutant for of the IP_3_ receptor to interfere with its function. Physiologically, the IP_3_ receptor-dependent impairment in the function of the dopaminergic neurons affects both synaptic vesicle release and excitability, Also, muscarinic acetylcholine receptors are required for proper development of the flight-modulating circuit during development.

The role of dopamine in the brain of *Drosophila* (as a model for general dopamine and brain function) is in the center of current research, and is studied by a large number of laboratories. More and more types of behavior are discovered that are modulated by dopaminergic neurons, and in particular those innervating the mushroom body. Therefore, the study is of very high interest for researchers working on *Drosophila*, but also to a broader readership.

The experiments are well designed. with appropriate controls at place. The conclusions drawn are highly interesting and novel (dopaminergic modulation of flight behavior, perhaps in the context of food seeking behavior, molecular mechanisms of circuit maturation).

Reviewer #2:

The results of the individual experiments reported by the authors are convincing. The approach is rigorous and they take full advantage of the many powerful molecular genetic tools available in *Drosophila*. The identification of a mechanism by which a small subset of dopaminergic cells may control behavior is significant.

Reviewer #3:

General Assessment: This study demonstrates that IP_3_R signaling (triggered by muscarinic receptor activation) affects excitability and quantal content of a subset of dopaminergic neurons to modulate flight duration and food search. I had no technical concerns and am generally supportive. My only major concern was that the narrative was fragmented. I believe this is because the perspective shifted between the IP_3_Rs and the dopamine neurons themselves, and was too focused. I think that streamlining the narrative and providing a broader perspective for the results will remedy this issue.

– I would like the authors to expand upon their final section of the Discussion to discuss more about 1) the potential context for cholinergic modulation of the PPL1-γ2α'1 DANs, 2) the proposed role of these DANs (which have been studied in several contexts) and 3) modulation of innate behavior in general. The paper begins with the importance of modulating innate behavior, but the discussion on this topic is spare and focused almost entirely on research on the mushroom bodies of *Drosophila*. The Discussion section, leans heavily on summarizing the results, rather than making connections to work in other systems or networks.

– The developmental section seemed somewhat tangential as the authors cannot distinguish between a developmental role for the IP_3_R from a need to express the *Itpr^DN^* transgene prior to adulthood to overcome a potential slow turnover of endogenous IP_3_R. In essence, it was unclear how these results contributed to the overall narrative of state modulation of behavior. Is this section informative to the development of the mushroom bodies or rigorous validation of the novel transgene?

---

## [Author Response]

Essential revisions:While there was considerable enthusiasm for the paper, there were concerns. Three particularly important concerns emerge from this review and subsequent discussion among the reviewers.1) There were concerns about the normality tests and reanalysis to avoid pseudo-replication that must be addressed.

We have now checked the data by two tests for normal distribution (Shapiro-Wilk and Kolmogorov_Smirnoff) and found that flight data do not follow a normal distribution. Therefore statistical analysis of flight data have now been performed using non-parametric tests. We have used the Kruskal-Wallis test followed by Dunn’s multiple comparison test for multiple comparisons and Mann-Whitney U-Test for pair wise comparisons. This information has been included in the statistical tests section in the Materials and methods. Regarding pseudo-replication, as suggested imaging data have been replotted and calculated now to include just one cell, or one lobe per brain. In addition we have included individual brain traces for every experiment as supplementary data (Figure 5—source data 1, Figure 6—source data 1, 2 and 3).

2) The Discussion should be made clearer and expanded to encompass more of the literature. Specifically, the authors should expand upon the final section of the Discussion to discuss more about 1) the potential context for cholinergic modulation of the PPL1-γ2α'1 DANs (For example, consider where the acetylcholine signal onto DANs might come from. DANs may not be entirely presynaptic to Kenyon cells but might also receive input from Kenyon cells.), 2) the proposed role of these DANs (which have been studied in several contexts) and 3) modulation of innate behavior in general. The paper begins with the importance of modulating innate behavior, but the discussion on this topic is spare and focused almost entirely on research on the mushroom bodies of *Drosophila.* The Discussion section leans heavily on summarizing the results, rather than making connections to work in other systems or networks.

As suggested we have now addressed each of these points in greater detail in the last section of the Discussion which has been expanded to two paragraphs. The possibility of cholinergic inputs from KC cells to DANs stimulating the IP_3_R have been included in the Discussion and in the final model in Figure 7. Several other references that mention the role of PPL1-γ2α'1 DANs in modulation of behaviour are now included – see last paragraph of the Discussion. We have expanded the last section of the Discussion to include possible roles for other regions of the brain in modulating flight and references to other insect brains, where relevant.

3) One common point raised by all reviewers was the need for expression of the itpr^DN^ during pupation which could have been due to either the perdurance of endogenous itpr vs. a developmental effect caused by the itpr^DN^ (the authors fully acknowledge the issue). This section raised many questions that aren't within the scope of this study, nor are easily resolved. Nevertheless, the authors must expand upon the implications of these results and suggest future studies will be needed to resolve the issue.

We are indeed unable to state equivocally if adult behavioural phenotypes, arising from expression of the IP_3_R^DN^, are only pupal or both pupal and adult. We have expanded on the implications of these results both in the Results (subsection “The IP_3_R affects neurotransmitter release from adult dopaminergic neurons”) and in theDiscussion (subsection “The IP_3_R and synaptic release in pupae and adults”). One way of addressing this is to express a tagged IP_3_R^DN^ specifically in late pupae and then follow its perdurance in adults. This experiment has now been suggested as a way to resolve this issue in the second paragraph of the Discussion.

Reviewer #3:General Assessment: This study demonstrates that IP_3_R signaling (triggered by muscarinic receptor activation) affects excitability and quantal content of a subset of dopaminergic neurons to modulate flight duration and food search. I had no technical concerns and am generally supportive. My only major concern was that the narrative was fragmented. I believe this is because the perspective shifted between the IP_3_Rs and the dopamine neurons themselves, and was too focused. I think that streamlining the narrative and providing a broader perspective for the results will remedy this issue.– I would like the authors to expand upon their final section of the Discussion to discuss more about 1) the potential context for cholinergic modulation of the PPL1-γ2α'1 DANs, 2) the proposed role of these DANs (which have been studied in several contexts) and 3) modulation of innate behavior in general. The paper begins with the importance of modulating innate behavior, but the discussion on this topic is spare and focused almost entirely on research on the mushroom bodies of *Drosophila*. The Discussion section, leans heavily on summarizing the results, rather than making connections to work in other systems or networks.

We have expanded the last section of the Discussion to include these suggestions (see above under consolidated review points).

– The developmental section seemed somewhat tangential as the authors cannot distinguish between a developmental role for the IP_3_R from a need to express the Itpr^DN^ transgene prior to adulthood to overcome a potential slow turnover of endogenous IP_3_R. In essence, it was unclear how these results contributed to the overall narrative of state modulation of behavior. Is this section informative to the development of the mushroom bodies or rigorous validation of the novel transgene?

The manuscript addresses how IP_3_R function impacts behaviour. In that context pupal (developmental) and adult contributions are both relevant.